# A Transparent Electrode Based on Solution-Processed ZnO for Organic Optoelectronic Devices

Zhi Chen[1], Jie Wang[1], Hongbo Wu[1], Jianming Yang[2], Yikai Wang[3], Jing Zhang[3], Qinye Bao[2], Ming Wang [1], Zaifei Ma [1] ✉, Wolfgang Tress [4] ✉ & Zheng Tang [1] ✉

Achieving high-efficiency indium tin oxide (ITO)-free organic optoelectronic devices requires the development of high-conductivity and high-transparency materials for being used as the front electrode. Herein, sol-gel-grown zinc oxide (ZnO) films with high conductivity (460 S cm$^{-1}$) and low optical absorption losses in both visible and near-infrared (NIR) spectral regions are realized utilizing the persistent photoinduced doping effect. The origin of the increased conductivity after photo-doping is ascribed to selective trapping of photogenerated holes by oxygen vacancies at the surface of the ZnO film. Then, the conductivity of the sol-gel-grown ZnO is further increased by stacking the ZnO using a newly developed sequential deposition strategy. Finally, the stacked ZnO is used as the cathode to construct ITO-free organic solar cells, photodetectors, and light emitting diodes: The devices based on ZnO outperform those based on ITO, owing to the reduced surface recombination losses at the cathode/active layer interface, and the reduced parasitic absorption losses in the electrodes of the ZnO based devices.

Transparent electrodes in organic optoelectronic devices are often based on indium tin oxide (ITO), which is the most successful transparent conducting material used in both academia and industry[1–4]. However, owing to the increasing demand and price for the rare material indium, alternative transparent electrodes are urgently needed for the development of organic optoelectronic technology, especially organic photovoltaics[5,6]. A promising candidate, mostly used in the field of organic photovoltaics, is poly(3,4-ethylenedioxythiophene) (PEDOT)[7,8], a transparent conjugated polymer that can be made conductive by poly(styrenesulfonate) (PSS) doping. However, due to its acidic nature, PEDOT:PSS used as the electrode in optoelectronic devices often causes rapid degradation of the device performance[9,10]. Carbon nanotubes[11], metal nanowire networks[12], and graphene[13] are also transparent conductors, with the potential to be used in optoelectronic devices. However, due to the low conductivity, undesired topographic properties, complicated synthetic routes, or limited commercial availability, a wide-range application of these materials in devices is currently not possible[14,15]. More importantly, the near-infrared (NIR) absorption strength of most of these materials is high[16,17], which with the typical layer thicknesses required for sufficient conductance for optoelectronic devices (100–500 nm), can induce significant parasitic optical losses, and thus limit the device performance.

[1]State Key Laboratory for Modification of Chemical Fibers and Polymer Materials, Center for Advanced Low-dimension Materials, College of Materials Science and Engineering, Donghua University, Shanghai 201620, P. R. China. [2]Key Laboratory of Polar Materials and Devices, School of Physics and Electronic Science, East China Normal University, 200241 Shanghai, P.R. China. [3]School of Material Science & Engineering, National Experimental Demonstration Center for Materials Science and Engineering, Jiangsu Collaborative Innovation Center of Photovoltaic Science & Engineering, Changzhou University, Changzhou 213164 Jiangsu, China. [4]Institute of Computational Physics, Zurich University of Applied Sciences, Wildbachstr. 21, 8401 Winterthur, Switzerland. ✉e-mail: mazaifei@dhu.edu.cn; wolfgang.tress@zhaw.ch; ztang@dhu.edu.cn

Apart from ITO, further metal oxides with a large bandgap can be made conductive and optically transparent[18]. Among them, doped zinc oxide (ZnO) is highly promising for being used as a transparent electrode in devices, because it can be deposited from solution using sol-gel methods[16,19]. Several strategies, such as Al- or Ga-doping, have been introduced to increase the conductivity of ZnO to over $10^5$ S cm$^{-1}$[20–22]. However, similar to the problem with the commonly used transparent conducting materials, the optical transmittance of doped ZnO is generally low, especially in the NIR spectral range[14,23]. Besides, these metal dopants are highly diffusive:[14] The diffusion of the dopants into the semiconducting active layer of optoelectronic devices severely deteriorates the device performance.

In the absence of intentional doping, sol-gel-grown ZnO is an n-type semiconductor with an extremely low absorption coefficient in both the visible and the NIR spectral region[18,23]. Thus, sol-gel-grown ZnO is commonly used as an electron-collecting interlayer between active material and cathode in optoelectronic devices[21,24–26]. The origin of the n-type conductivity in the sol-gel-grown ZnO is often ascribed to oxygen vacancies ($V_O$) in the ZnO lattice[27–30], although whether $V_O$ is a shallow donor state or a deep trap state is still being heavily debated[31]. Besides, impurities with low formation energies, including interstitial hydrogen ($H_i$) and substitutional hydrogen ($H_O$), are the most frequently discussed origins for the n-type conductivity in ZnO[29,31,32]. Nevertheless, despite being an n-type semiconductor, sol-gel-grown ZnO has very low conductivity[28,30,33], on the order of 1–10 S cm$^{-1}$. Therefore, so far it is not suitable as an electrode material for optoelectronic devices.

In this work, we propose a strategy to increase the conductivity, while maintaining the high optical transparency of sol-gel-grown ZnO thin films, and realize a successful replacement of ITO in high-efficiency optoelectronic devices. In contrast to the general approach to increase the conductivity of ZnO, i.e., increasing charge carrier mobilities or the degree of intentional doping, the strategy developed here is based on the photoinduced doping effect. Accordingly, we realize record-high conductivity of close to 500 S cm$^{-1}$ and ultra-low optical absorption losses (<1%, for the wavelengths longer than 400 nm) for our ZnO films. More specifically, we demonstrate that the high conductivity of ZnO is a result of the combined efforts of 1) intensive thermal treatment, reducing the concentration of organic residuals in the ZnO film, 2) insertion of layer interfaces, increasing the density of $V_O$ that act as hole scavengers in the ZnO film, and 3)

UV light-induced persistent doping, providing excess long-living electrons in the conduction band (CB) of the ZnO film. Subsequently, we demonstrate organic solar cells, photodetectors, and light-emitting diodes based on a ZnO cathode, and realize device performance better than that of the devices based on ITO. This marks a milestone in the development of alternative transparent electrodes to ITO for optoelectronic devices.

## Results

### Conductive and transparent ZnO thin films grown using the sol-gel method

First, ZnO thin films were grown using the typical sol-gel method reported in the literature[19]. The precursor solution, prepared by mixing zinc acetate, ethanolamine, and 2-methoxyethanol, is spin-cast onto a glass substrate, then the substrate is annealed, under a controlled relative humidity of 20%. The annealing temperature used is 160 °C, and the thickness of the final ZnO film is about 20–30 nm. As expected, the conductivity of the ZnO grown using the sol-gel method is extremely low: The sheet resistance of the ZnO film is beyond the measurement range of the four-point probe setup used in this work (284 Mohm). Thus, the sol-gel-grown ZnO could not be used as the cathode for optoelectronic devices.

Generally, the conductivity of ZnO increases after UV treatment, due to the UV-induced doping effect:[34,35] The UV-generated holes are trapped by defects or impurities in the ZnO lattice, leaving free excess electrons in the CB of ZnO[31,36]. We thus expect a similar increase in conductivity for the sol-gel-grown ZnO after UV treatment. Indeed, as shown in Fig. 1a, we find that UV treatment has a significant impact on the conductivity of the sol-gel-grown ZnO film: After UV treatment (365 nm, 24 W, and 1000 s), the conductivity of the ZnO film is increased to 0.3 S cm$^{-1}$, orders of magnitude higher than that of the non-treated ZnO.

Several mechanisms have been proposed to explain the selective hole trapping process[28,37–40]. The one expected to be most relevant for the sol-gel-grown ZnO is the existence of neutral $V_O$ sites with an energy level close to the valence band (VB) maximum of ZnO[34,40]. This is later confirmed in this work by photoluminescence (PL) and spectroscopic ellipsometry measurements. The energetic diagram illustrating the hole trapping process by $V_O$ in ZnO is shown in Fig. 1b. Note that the increased conductivity in the sol-gel-grown ZnO due to the trapping of UV-generated holes is found to be persistent, i.e., the

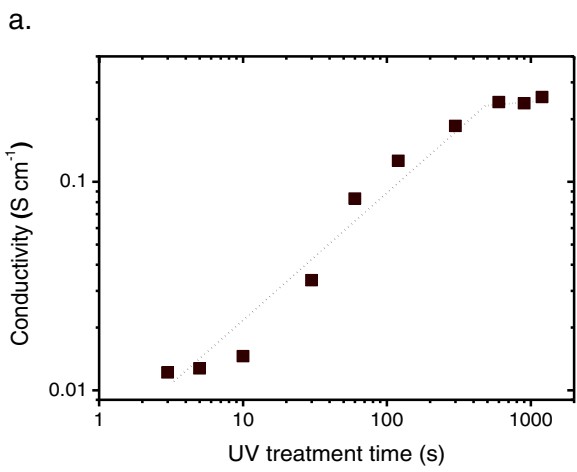
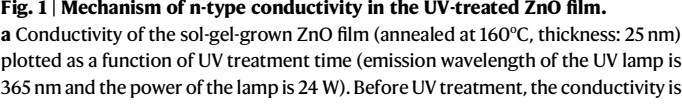
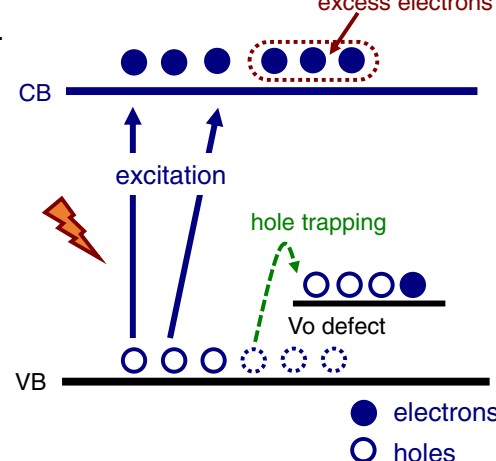

**Fig. 1 | Mechanism of n-type conductivity in the UV-treated ZnO film.**
**a** Conductivity of the sol-gel-grown ZnO film (annealed at 160ºC, thickness: 25 nm) plotted as a function of UV treatment time (emission wavelength of the UV lamp is 365 nm and the power of the lamp is 24 W). Before UV treatment, the conductivity is too low to be measurable by the four-point probe measurement setup used in this work. **b** Schematic representation for the selective trapping of the UV-generated holes by $V_O$ in ZnO. The trapping of holes leaves excess electrons in the CB, increasing the conductivity of ZnO.

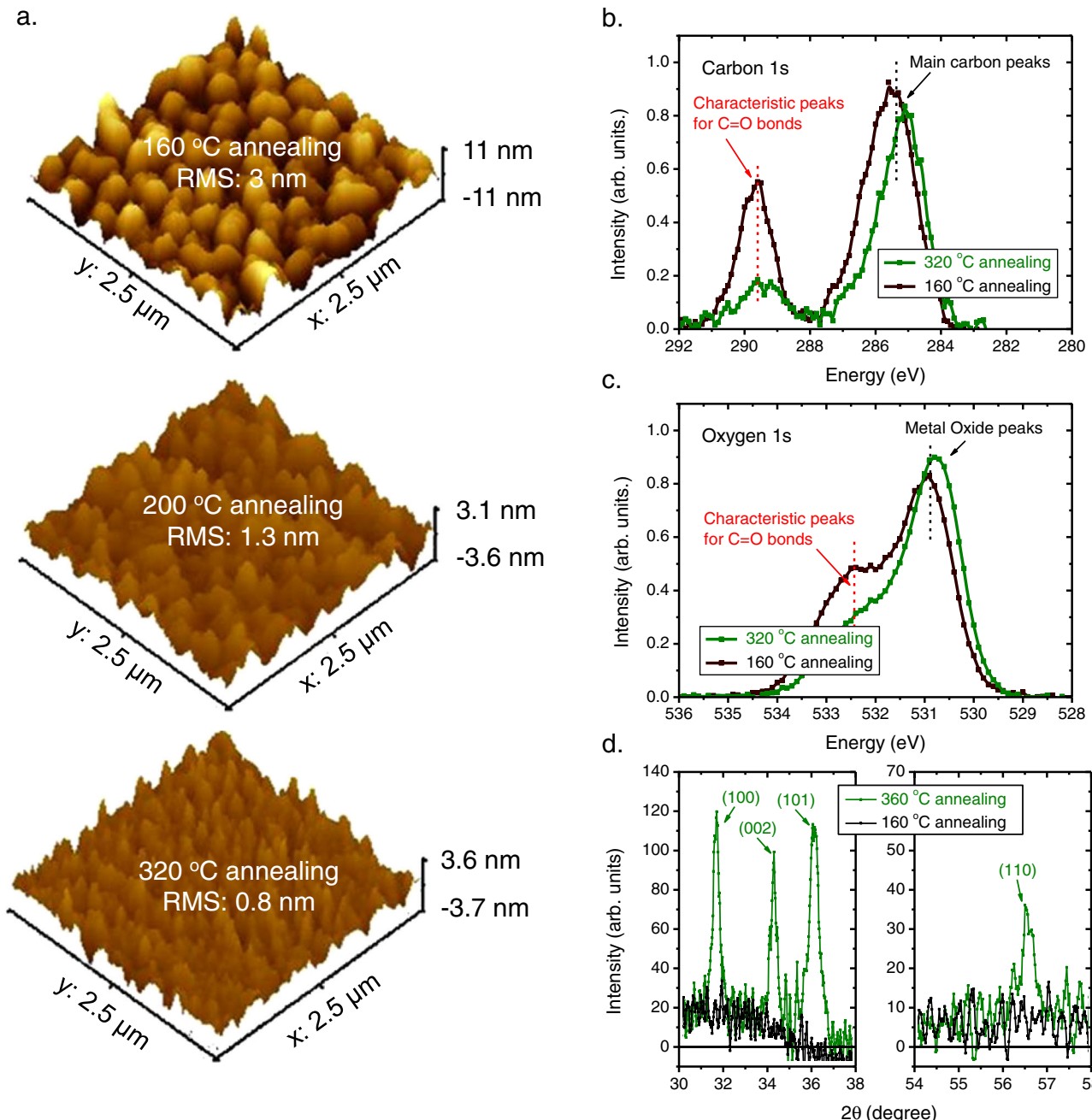

**Fig. 2 | Structural and compositional properties of the ZnO films annealed at different temperatures. a** AFM images of the sol-gel-grown ZnO film annealed at different temperatures. **b** XPS carbon 1 s spectra and **c** XPS oxygen 1 s spectra for the ZnO film annealed at different temperatures. **d** XRD spectra of the ZnO film annealed at different temperatures.

high conductivity remains after the removal of UV illumination, which is similar to that reported in the literature[36]. This could be due to the very slow kinetics for the recombination of the trapped holes (with the free electrons in the CB of ZnO), in comparison to the rate of photo-excitation of the subgap transition (for instance, excitation of electrons from Vo to CB) enabled by ambient light conditions, as discussed in the literature[35]. This effect will also be discussed in more detail later in this article.

Nevertheless, the conductivity of the UV-treated ZnO thin film is still low, likely due to the incomplete formation of ZnO nanocrystals in the sol-gel-grown thin film[19,41]. To solve this problem, we employed a more intensive thermal treatment for growing ZnO, and we find that a higher annealing temperature alters both the structural and the compositional properties of ZnO. More specifically, atomic force

microscopy (AFM) and scanning electron microscope (SEM) measurements reveal a significant difference in the topographic properties of the ZnO films annealed with different temperatures, as shown in Fig. 2a and Supplementary Fig. 1, the surface of ZnO annealed at 160°C is rough, with island structures and the diameter of the islands being about a few hundred nanometers. The root-mean-square (RMS) surface roughness value is 3 nm, indicating phase separation and an incomplete decomposition of the organic components in the low-temperature processed ZnO film[19]. On the other hand, the surfaces of the ZnO films annealed at higher temperatures are much smoother, with RMS values smaller than 1 nm. The X-ray photoelectron spectroscopy (XPS) measurements (Fig. 2b) reveal that the characteristic carbon 1 s peak for the C = O bonds (289.6 eV), originating from the organic residuals, is much weaker in the ZnO film grown with a higher

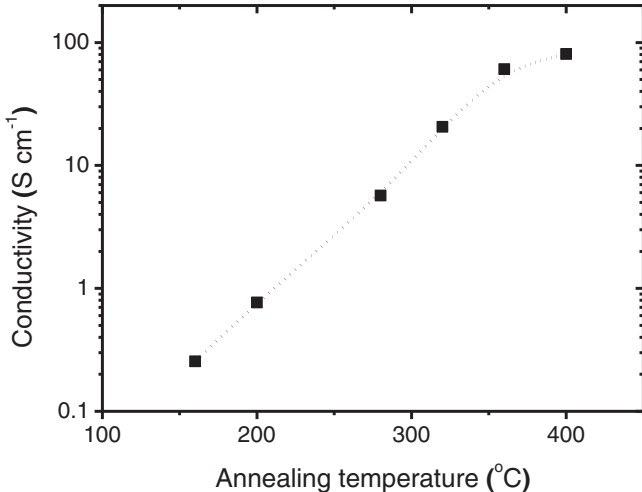

**Fig. 3 | Impact of UV treatment on conductivity of the ZnO films annealed at different temperatures.** The conductivity of the UV-treated (365 nm, 24 W, and 600 s) ZnO film annealed with a higher temperature is higher than that annealed with a lower temperature.

annealing temperature. This suggests that the degree of decomposition of the organic component increases with the annealing temperature. Furthermore, from the oxygen 1 s peaks, we also find that the intensity of the C=O peak (532.5 eV) is much lower for the ZnO film annealed with a higher temperature (Fig. 2c), suggesting a higher density of ZnO nanocrystals in the film.

X-ray diffraction (XRD) measurements are also performed for the ZnO films annealed with different temperatures. As shown in Fig. 2d, the XRD patterns of the ZnO film annealed with a higher temperature possess a polycrystalline hexagonal wurtzite structure with no preferred orientation. However, for the ZnO film annealed with a lower temperature, the diffraction peaks are non-detectable, suggesting that the formation of ZnO nanocrystals in the film annealed with a lower temperature is far from being complete. The XRD results again confirm that increasing the annealing temperature is beneficial for the formation of ZnO nanocrystals, and thus, increasing the conductivity of the UV-treated ZnO film.

Subsequently, we achieve significantly improved conductivity in the UV-treated ZnO annealed at a higher temperature (Fig. 3). The conductivity of the ZnO film annealed at 400°C is almost 100 S cm$^{-1}$, which is already close to the highest value reported for sol-gel-grown ZnO[36]. Also, we find that the UV treatment does not alter the structural or compositional properties of ZnO: The XPS and the XRD spectra (Supplementary Fig. 2) are the same and the topographic properties, according to the AFM and the SEM images (Supplementary Fig. 3), are similar for the ZnO film before and after UV treatment. This suggests that the increased conductivity of ZnO after UV treatment is most likely a result of increased charge carrier density (due to the trapping of photo-generated holes). Note that an accurate determination of the charge carrier concentration in the sol-gel-grown ZnO using the Hall effect measurement is found to be highly challenging, possibly due to the thinness of the film (20–30 nm), or the fact that the hole trapping is a surface effect, difficult to probe with the Hall effect measurement[42]. This will be further discussed in the next section of this article.

## Increasing the conductivity of ZnO using the sequential deposition strategy

For the sol-gel-grown ZnO to act as a cathode in organic optoelectronic devices, the conductivity of ZnO needs to be further increased. This requires increasing the density of the hole traps in the ZnO film. In the sol-gel-grown ZnO film, the introduction of hole traps including $V_O$ and hydroxyl groups, or carbon contamination[43–45], etc., is primarily

assigned to the adsorption of gas molecules at the ZnO surface[46]. For instance, adsorbed water molecules have been observed to dissociate into OH$^-$ and H$^+$ at the film surface[47,48]. Subsequently, H$^+$ reacts with the oxygen atoms in the ZnO lattice, leading to a depletion of oxygen at the ZnO surface. This results in a high density of $V_O$ at the ZnO surface[49,50], and a high capability of the ZnO surface to trap photo-generated holes. In principle, the degree of water molecule adsorption at the ZnO surface could be increased via increasing the environmental humidity or increasing the surface roughness (i.e., the specific surface area) of the ZnO thin film. However, this is non-trivial, since the n-type conductivity of the sol-gel-grown ZnO is strongly dependent on the processing conditions, such as precursor concentration, spin-coating speed, environmental humidity, etc.[51,52]. The conductivity of ZnO is easily compromised by the deposition process deviating from the already optimized processing condition.

In this regard, we develop a sequential deposition method for the growth of the ZnO stack, as schematically illustrated in Fig. 4a: Using the sequential deposition method, multiple layer interfaces, capable of trapping water, are introduced into the ZnO stack, without the need to alter the processing condition for the deposition of the ZnO layer. Therefore, the density of $V_O$, able to trap photogenerated holes, is expected to be higher in the ZnO stack, compared to that in the single-layer ZnO. Accordingly, the conductivity of the ZnO stack is expected to be higher.

To confirm that the density of $V_O$ in the sol-gel-grown ZnO could indeed be increased by using the sequential deposition strategy, PL measurements are performed for the ZnO stack with different numbers of ZnO layers, annealed at 320 °C, and the results are shown in Fig. 4b. For the single-layer ZnO, the highest peak, assigned to the band-to-band transition of free charge carriers, is located at 380 nm (3.3 eV)[53]. For the ZnO stacks, the highest emission peak red-shifts with an increasing number of ZnO layers (from 380 to 400 nm). This suggests a shift from band-to-band to excitonic transitions[31]. The excitonic state is mostly found in oxygen-deficient materials[31]. Thus, the shift of the PL emission peak suggests that the oxygen content decreases, and thereby the density of $V_O$ increases with the number of ZnO layers in the stack. Furthermore, two additional emission peaks are identified in the PL spectra of the ZnO stacks. The first one is at 480 nm (2.6 eV), frequently assigned to the transition of electrons recaptured by the ionized $V_O$ state to the VB[54]. From the PL spectra, it is clear that the ratio between the intensities of the peak at 480 nm and the peak at 380–400 nm increases with the number of ZnO layers. This also suggests that the density of $V_O$ increases. The second narrow emission peak is at 530 nm (2.3 eV), corresponding to the transition of electrons from the CB to the neutral $V_O$ states[55], a transition that is made possible when the neutral $V_O$ states are filled with trapped holes. This confirms that the UV-generated holes could be trapped by $V_O$ in the sol-gel-grown ZnO. Importantly, it is noted that the intensity ratio between the peak at 530 nm and the peak at 380–400 nm increases with the number of ZnO layers, again confirming the density of the trapped holes, and thus, the density of $V_O$, increases. Therefore, the conductivity of the ZnO stack should increase with the number of ZnO layers. The schematic picture illustrating the energy and the wavelength of the transitions revealed in the PL is shown in Fig. 4c. It should be noted that the transition probability of electrons from the CB to the hole-filled neutral $V_O$ state should be extremely low, to allow for the realization of the persistent high conductivity of ZnO after UV treatment. Nevertheless, we are able to observe the characteristic PL emission from this transition, because we use a high-intensity UV laser excitation (320 nm, 150 W).

Then we perform the UV treatment for the ZnO stacks and determine the electrical properties of the ZnO stacks with different numbers of ZnO layers. From the four-point probe measurements (Fig. 4d), we find that the conductivity of the UV-treated ZnO stack indeed increases with the number of ZnO layers. An exception is found

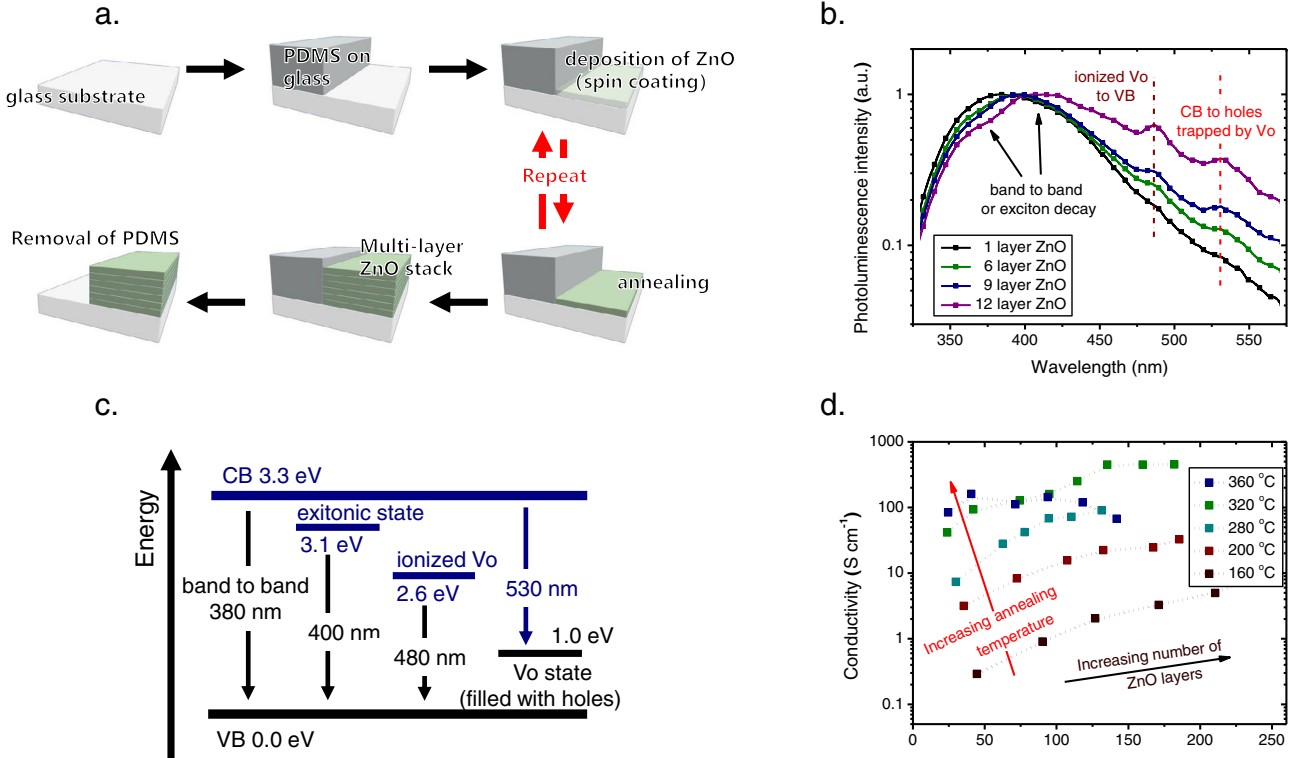

**Fig. 4 | ZnO grown using the sequential deposition strategy. a** Schematic representation of the sequential deposition strategy for the growth of the ZnO stack. The use of polydimethylsiloxane (PDMS) is for the patterning of the ZnO stack, later to be used for constructing optoelectronic devices. **b** PL spectra of the ZnO stacks with different numbers of ZnO layers, annealed at 320 °C (excitation wavelength 320 nm, 150 W). **c** Energy diagram for the different transitions in the ZnO stack, identified by the PL measurements. **d** Conductivity of the ZnO stacks with different numbers of ZnO layers, annealed with different temperatures, measured after UV treatment. The deposition and annealing of the wet ZnO films are done in a controlled environment with relative humidity being 20%.

for the ZnO stack annealed at a very high temperature of 360 °C: A declining trend in conductivity is observed with the increasing number of ZnO layers. This could be due to a structural change of ZnO under the repeated intensive thermal treatments, according to the SEM and XRD measurements (Supplementary Fig. 4)[41]. For the ZnO annealed at a slightly lower temperature, i.e., 320 °C, the conductivity is increased from 40 S cm$^{-1}$ for the single-layer ZnO to 100 S cm$^{-1}$ for the ZnO stack with 2 layers of ZnO, and to a remarkable 460 S cm$^{-1}$ for the ZnO stack with 6 layers of ZnO. We note that the conductance of the ZnO stack continues to increase with the number of ZnO layers, although the conductivity does not further increase.

From the ultra-violet photoemission spectroscopy (UPS) measurements (Fig. 5a, b), we note that the work function of the ZnO stack is about 4.0 eV, significantly lower than that of ITO (4.5 eV). This suggests that the ZnO stack developed in this work is more suitable than ITO, as the cathode of organic optoelectronic devices. However, a work function of 4.0 eV could still be too high for realizing an ohmic electron contact for optoelectronic devices. For instance, for solar cells based on the most efficient acceptor materials, including 2,2′-((2Z,2′Z)-((12,13-bis(2-ethylhexyl)−3,9-diundecyl-12,13-dihydro-[1,2,5] thiadiazolo[3,4-e]thieno[2″,3″:4′,5′]thieno [2′,3′:4,5]pyrrolo[3,2-g] thieno[2′,3′:4,5]thieno[3,2-b]indole-2,10-diyl)bis (methanylylidene)) bis(5,6-difluoro-3-oxo-2,3-dihydro-1H-indene-2,1-diylidene)) dimalo-nonitrile (Y6)[56] or 3,9-bis(2-methylene-(3-(1,1-dicyanomethylene)-indanone))−5,5,11,11-tetrakis(4-hexylphenyl)-dithieno[2,3-d:2′,3′-d′]-s-indaceno[1,2-b:5,6-b′] dithiophene (ITIC)[57], the lowest-unoccupied-molecular-orbital (LUMO) energy level of the active layer lays at approximately 3.9 eV, thus, the desired work function of the cathode should be lower than 3.9 eV. Nevertheless, the problem of having a too high work function for the ZnO stack could, in principle, be easily circumvented by employing a cathode interlayer, such as

polyethyleneimine ethoxylated (PEIE)[58], poly(3,3′-(((9′,9′-dioctyl-9H,9′H-[2,2′-bifluorene]−9,9-diyl)bis(4,1-phenylene)) bis(oxy))bis(N,N-dimethylpropan-1-amine)) (PFPA-1)[59], or poly(9,9-bis(3′-(N,N-dimethyl)-N-ethylammoinium-propyl-2,7-fluorene)-alt-2,7-(9,9-dioctyl-fluorene)) dibromide (PFN-Br)[60], etc., to modify the work function of the sol-gel-grown ZnO, as will be demonstrated in the next section of this article.

Spectroscopic ellipsometry measurements are performed to characterize the optical properties of the ZnO stack grown using the sequential deposition method. The absorption coefficient spectra of the ZnO stack with 6 layers of ZnO, annealed at 320 °C, before and after UV treatment are shown in Fig. 5c. We observe the primary ZnO absorption peak at 3.5 eV with an absorption coefficient being the same for both the UV-treated and non-treated ZnO films. We also note the rather strong subgap absorption band for the ZnO stack, peaking at 2.6 eV. This absorption feature is typical in ZnO with a high density of $V_O$[61], corresponding to the excitation kinetics of the electrons from the neutral $V_O$ states to the CB. Therefore, the reduced strength of the subgap absorption for the ZnO stack after the UV treatment could be ascribed to the trapping of the UV-generated holes by the neutral $V_O$ states: The trapping of holes reduces the density of electrons on the neutral $V_O$ states, thus, reduces the probability of the excitation of the electrons on the $V_O$ states. Moreover, from Fig. 5d, we find that the optical transparency of the sequentially deposited ZnO stack is very high: The absorption coefficients (in both the visible and the NIR spectral range) of the ZnO stack are orders of magnitude lower than those of the common transparent conductors, such as PEDOT:PSS, Al-doped ZnO, and ITO. Thus, the transmittance of the ZnO stack is much higher, as shown in Fig. 5e.

We also find that the conductivity of the air-exposed ZnO stack slowly reduces after the removal of UV illumination: The conductivity

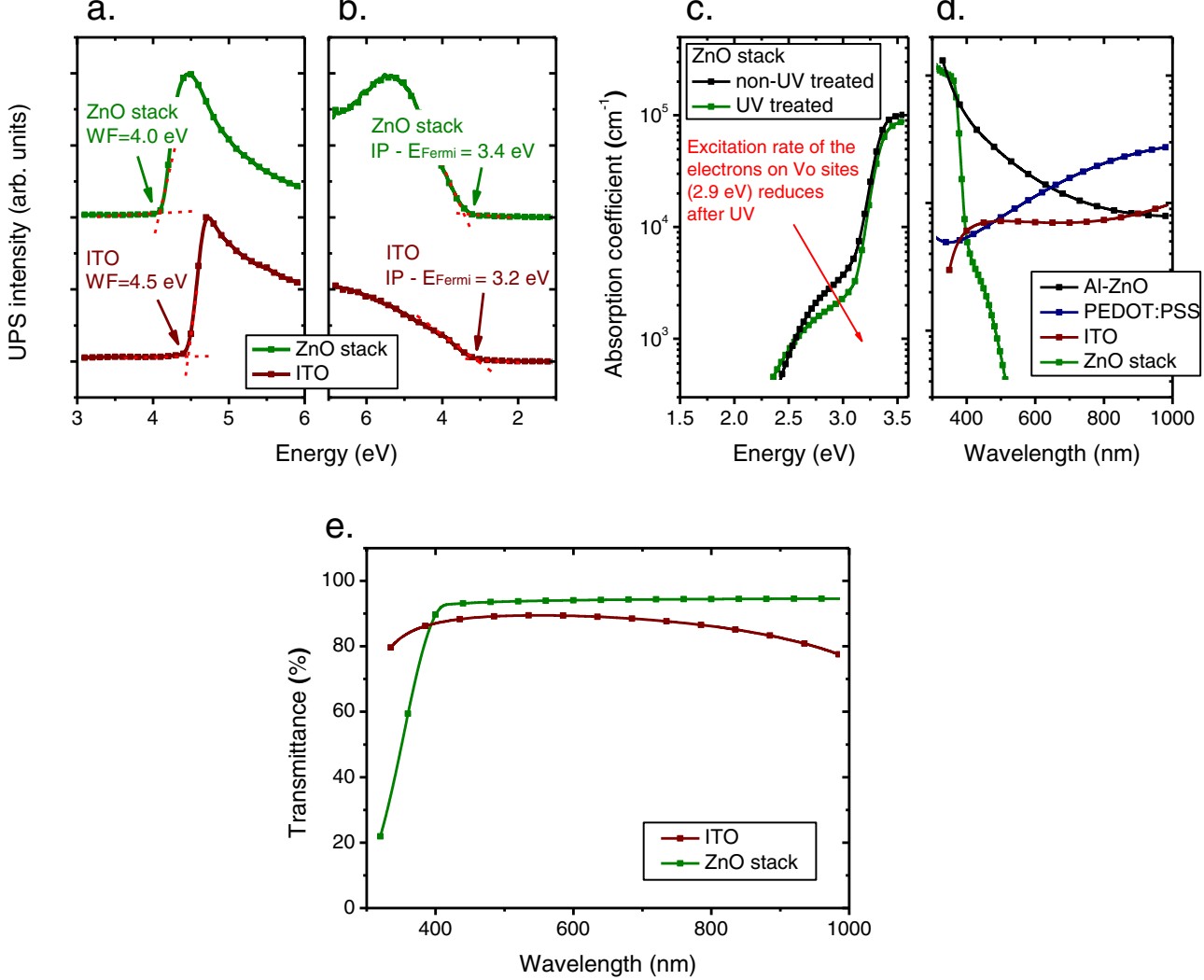

**Fig. 5 | Photoelectrical and optical properties of the ZnO stack. a** Work function (WF) and **b** ionization potential of ITO and the ZnO stack with 6 layers of ZnO annealed at 320 °C, deposited using the sequential deposition strategy. **c** Absorption coefficient of the ZnO stack before and after UV treatment (365 nm, 24 W, 600 s), measured by spectroscopic ellipsometry. **d** Absorption coefficient of the ZnO stack compared to that of the films based on Al-doped ZnO, PEDOT:PSS, and ITO. **e** Transmittance spectra of the ZnO stack (150 nm) and the ITO film (170 nm) were used in this work.

of the UV-treated ZnO stack with 6 ZnO layers stored under ambient conditions is reduced from 460 S cm$^{-1}$ to 160 S cm$^{-1}$, 7 h after the UV treatment (Supplementary Fig. 5a). This is not unexpected, since the hollow sites at the hexagon centers of the ZnO nanocrystals are capable of adsorbing oxygen molecules: After oxygen adsorption, the free electrons in the CB of the ZnO nanocrystals are transferred to the adsorbed oxygen molecules, leading to the formation of Zn-O bonds and the reduction of the density of free electrons and Vo sites, resulting in the increase of resistance of the ZnO film[62]. Meanwhile, air exposure of the UV-treated ZnO also leads to water adsorption on the ZnO surface. Because the adsorbed water molecules are able to react with the newly formed Zn-O bonds, breaking the Zn-O bonds, as discussed before, leads to the increase of density of Vo sites in the ZnO that could be excited again under UV illumination. Thus, the reduced conductivity is easily recovered by a second UV treatment, as shown in Supplementary Fig. 5a.

Note that since the decrease of conductivity of the UV-treated ZnO stack is induced by air exposure, it should not cause stability issues for the devices with the ZnO stack acting as the transparent electrode, because the UV treatment could be performed after the completion of device construction and encapsulation. In fact, we find that the stability of solar cells based on the ZnO stack is generally

better than that based on ITO, which will be demonstrated later in this article.

## ITO-free organic solar cells based on the ZnO stack

We now investigate the performance of organic solar cells based on the ZnO stack. The device architecture is shown in Fig. 6a. The ZnO stack acting as the bottom cathode is deposited to cover only half of the glass substrate, and PEIE is deposited on top of the ZnO stack to modify the cathode work function. The top MoO$_3$/Ag electrodes with a stripe pattern are deposited via thermal evaporation. The reference solar cell is based on the typical inverted device architecture of ITO/ZnO/active layer/MoO$_3$/Ag, with the single-layer ZnO interlayer grown using the standard method reported in the literature[19]. The active area of the solar cells is about 5 mm$^2$.

For better reproducibility, the ZnO stack with 6 ZnO layers is used as the bottom transparent cathode, and the annealing temperature used for growing ZnO is 320 °C. First, a blend of poly[(2,6-(4,8-bis(5-(2-ethylhexyl)thiophen-2-yl)-benzo[1,2-b:4,5-b']dithiophene))-alt-(5,5-(1',3'-di-2-thienyl-5',7'-bis(2-ethylhexyl)benzo[1',2'-c:4',5'-c']dithiophene-4,8-dione)] (PBDB-T):ITIC[63,64] (weight ratio, 1:1) is used as the photoactive material system for constructing the solar cell. The completed device is UV-treated (365 nm, 24 W,

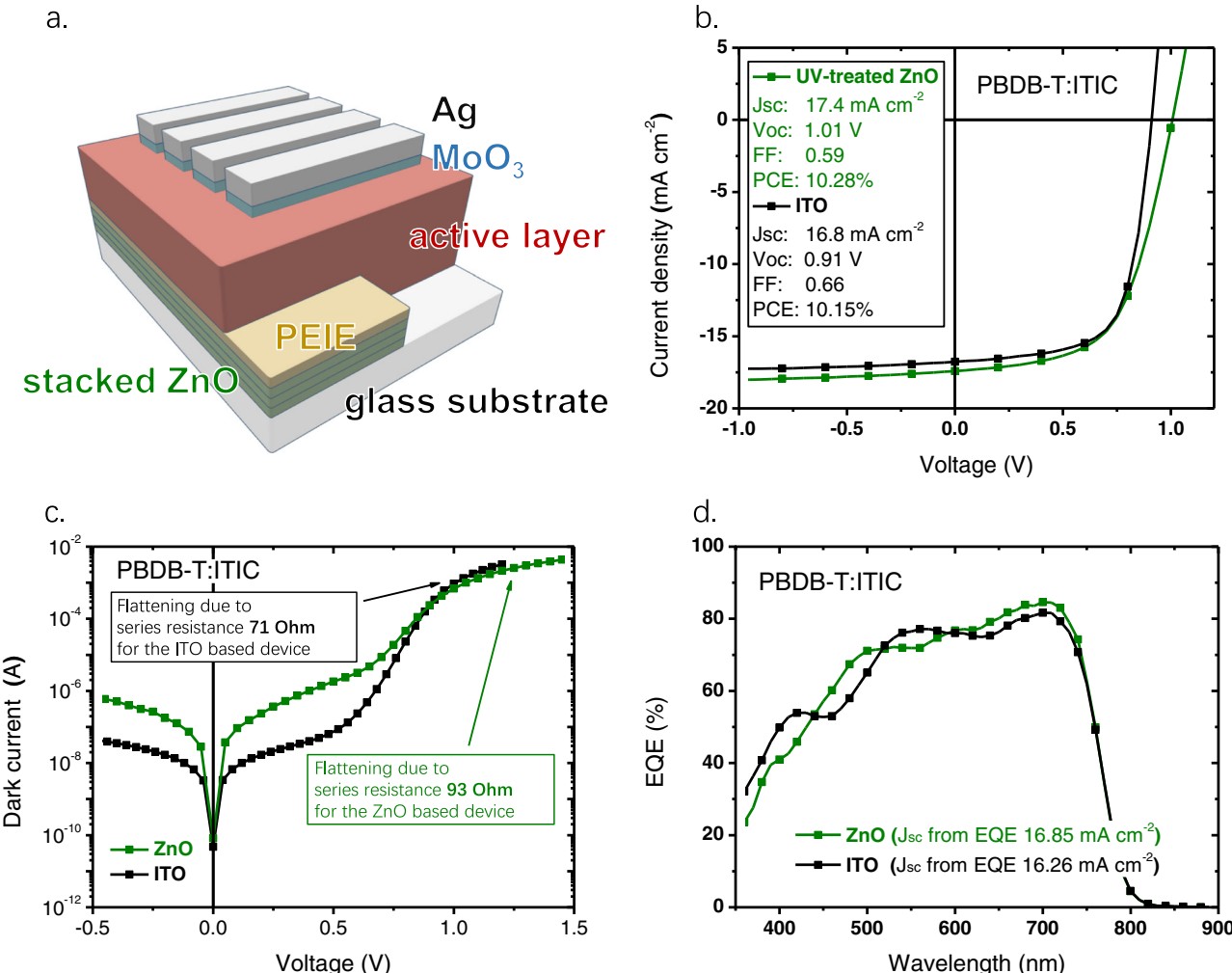

**Fig. 6 | ITO-free solar cells based on the ZnO stack. a** Schematic device architecture for the ITO-free solar cells based on the ZnO stack acting as the cathode. **b** Light *J-V* curves, **c** dark current-voltage curves, and **d** EQE spectra of the PBDB-T:ITIC solar cells based on ITO and the ZnO stack with 6 layers of ZnO, annealed at 320 °C. The solar cell based on ZnO is UV-treated (365 nm, 24 W, and 600 s), prior to characterizing its performance, and the solar cell based on ITO is not UV-treated. A comparison between the performance of the UV-treated solar cells based on ZnO and ITO is provided in Supplementary Fig. 6.

600 s), prior to characterizing its performance. The current density-voltage (*J-V*) characteristic curves of the solar cells based on ZnO and ITO are shown in Fig. 6b. From the *J-V* curves, we find that the short-circuit current density ($J_{sc}$) of the solar cell based on ZnO is similar to that based on ITO, suggesting that the series resistance of the ZnO cathode is sufficiently low to not limit photocurrent under short circuit. Then, we determine the series resistance of the solar cells based on ZnO and ITO from the dark *J-V* curves. As shown in Fig. 6c, we find that the device series resistance of the ZnO based solar cell is 93 Ohm, slightly higher than that of the ITO based device (71 Ohm). Since the $J_{sc}$ values of the solar cells based on ZnO and ITO are similar, the peak external quantum efficiency (EQE) values of the solar cells based on ZnO and ITO are also similar (Fig. 6d). However, we note that the spectral shapes of EQE for the solar cells based on ZnO and ITO are slightly different, which could be ascribed to the different dielectric functions of ZnO and ITO, leading to different optical interference conditions in the devices. Strikingly, the open-circuit voltage ($V_{oc}$) of the PBDB-T:ITIC solar cell based on ZnO is high, over 1.0 V, which is a record value for PBDB-T:ITIC solar cells[64], whereas the $V_{oc}$ of the ITO based solar cell is only 0.91 V. This is not a result of photocharging in the active layer of the ZnO based solar cell, since the higher $V_{oc}$ is persistently obtained, and it does not depend on the direction of the voltage sweep used during the *J-V* measurements.

The $V_{oc}$ of an organic solar cell is primarily determined by the energy of the bandgap ($E_g$)[65] and the energy of charge transfer states ($E_{CT}$) of the active layer[66]. However, we find that the energetic properties of the active layer of the ZnO and the ITO-based solar cells are very similar, as listed in Table 1, and the main reason for the different $V_{oc}$ is associated with the reduction of the non-radiative decay rate of charge carriers in the ZnO based solar cell, as discussed in detail in Supplementary Note 1. Thus, we ascribe the increased $V_{oc}$ of the ZnO-based solar cell to the suppressed non-radiative decay channels in the active layer or the interfaces between the active layer and the electrodes.

Trap-assisted recombination and surface recombination are the frequently observed non-radiative decay channels in organic solar cells. For efficient organic active materials systems, such as the used PBDB-T:ITIC system, the density of traps is expected to be low. Indeed, light intensity-dependent *J-V* measurements reveal that the slope of the $V_{oc}$ *vs* light intensity curve (Supplementary Fig. 8) is about 1.3 *kT*, for both the ZnO and the ITO-based solar cells, suggesting weak trap-assisted recombination losses in these devices[67]. Thus, surface recombination is more likely the reason for the different $V_{oc}$ in the solar cells based on ZnO and ITO.

In organic solar cells, the surface recombination is a result of the diffusion of minority charge carriers to the contact for the majority charge carriers, or in other words, the loss of photogenerated

**Table 1 | The representative photovoltaic performance parameters and the voltage loss values from the EQE and EL measurements for the PBDB-T:ITIC solar cells based on ZnO and ITO electrode. The statistical results for the photovoltaic performance parameters are provided in Supplementary Table 1**

| | $J_{sc}$ (mA cm$^{-2}$) | $V_{oc}$ (V) | FF | PCE (%) | $E_g$ (eV) | $E_{CT}$ (eV) | $\Delta V_{loss}$[a] (V) | $J_{0,rad}$[b] (mA cm$^{-2}$) | $V_{oc,rad}$[c] (V) | $\Delta V_{nr}$[d] (V) | $\Delta V_r$[e] (V) | $EQE_{EL}$[f] (%) |
|---|---|---|---|---|---|---|---|---|---|---|---|---|
| ITO | 16.8 | 0.91 | 0.66 | 10.15 | 1.64 | 1.47 | 0.73 | $1.0 \times 10^{-20}$ | 1.22 | 0.31 | 0.42 | 0.02 |
| ZnO | 17.4 | 1.01 | 0.59 | 10.28 | 1.64 | 1.47 | 0.63 | $1.0 \times 10^{-20}$ | 1.22 | 0.21 | 0.42 | 0.11 |

[a]Total voltage losses, calculated using eq. $\Delta V_{loss} = E_g/q - V_{oc}$, where q is the elementary charge. The method used to determine $E_g$ is provided in Supplementary Fig. 7.
[b]Saturation current density, calculated from EQE.
[c]Radiative limit for $V_{oc}$, calculated from $J_{0,rad}$.
[d]Non-radiative recombination voltage loss, calculated using eq. $\Delta V_{nr} = V_{oc,rad} - V_{oc}$.
[e]Radiative recombination voltage loss, calculated using eq. $\Delta V_r = E_g/q - V_{oc,rad}$.
[f]The external quantum efficiency of electroluminescence.

electrons (holes) at the anode (cathode), driven by the concentration gradient of free charge carriers in the active layer. Suppressed diffusion loss can be a result of increased contact selectivity[68] or an increased built-in electric field[69]. Since we do not expect to have highly selective contacts in the solar cells based on either ZnO or ITO, we can only expect the latter, i.e., the increased built-in electric field, to be the explanation for the higher $V_{oc}$ of the ZnO-based solar cell. Indeed, the work function of the ZnO stack coated with PEIE is 3.7 eV (Supplementary Fig. 9), lower than that of the ITO coated with a single-layer ZnO (4.1 eV). This allows for the realization of a larger difference between the work functions of the cathode and the anode, and thus, a higher built-in electric field in the ZnO-based solar cell.

Then, we use the more efficient poly[(2,6-(4,8-bis(5-(2-ethylhexyl-3-fluoro)thiophen-2-yl)-benzo[1,2-b:4,5-b']dithiophene))-alt-(5,5-(1',3'-di-2-thienyl-5',7'-bis(2-ethylhexyl)benzo[1',2'-c:4',5'-c']dithiophene-4,8-dione)] (PM6):Y6 active material system[56] to construct ZnO based solar cells, and we find that the $V_{oc}$ of the ZnO based solar cell is 0.89 V (Supplementary Fig. 10), which is also a record value for PM6:Y6 solar cells. However, the fill-factor (FF) of the solar cell based on ZnO is lower, compared to that based on ITO, limiting the power conversion efficiency (PCE) of the solar cell. The limited PCE of the ZnO based solar cell is ascribed to the fact that we always perform the UV treatment after the completion of device construction and encapsulation, since the conductivity of the UV-treated non-encapsulated ZnO decreases over time. However, Y6 degrades rapidly under UV illumination[70]. Therefore, the results from the PM6:Y6 systems imply that our strategy of replacing ITO with ZnO is less efficient for the solar cells based on the active materials systems that degrade under UV illumination. Considering the fact that organic solar cells must be used under solar illumination (containing substantial UV photons), organic molecules that degrade under UV illumination would anyway not be suited for practical applications. Thus, the universality of our strategy should not be considered as compromised by the results from the PM6:Y6 based solar cells.

To demonstrate the universality of our strategy of replacing ITO with ZnO, we also include the experimental results from the ternary solar cells based on PM6: 3,9-bis(2-methylene-((3-(1,1-dicyano-methylene)−6,7-difluoro)-indanone)−5,5,11,11-tetrakis(4-hexylphenyl)-dithieno[2,3-d:2',3'-d']-s-indaceno[1,2-b:5,6-b']dithiophene (IT4F): [6,6]-Phenyl-C71-butyric acid methyl ester (PCBM)[71], another high-efficiency materials system, which is found to be more stable than the Y6 based active materials system, under UV illumination. As demonstrated in Supplementary Fig. 12, the ZnO based solar cell is more efficient than the ITO based solar cell, due to the higher $V_{oc}$, similar to the results obtained from the ZnO and the ITO based solar cells with the active layer based on the model PBDB-T:ITIC system.

Also, we show that the use of the ZnO stack as the cathode of organic solar cells does not lead to reduced long-term stability of the device: For all of the active materials systems used in this work, the PCE of the ZnO-based solar cells retained 70% of their initial value after one month of shelf storage (Supplementary Fig. 13), whereas the PCE of the

ITO based solar cells degraded to 70% during the course of two weeks. In addition, under continuous illumination (100 mW cm$^{-2}$), the stability of the ZnO based solar cells is found to be better than that of the ITO-based solar cells, as shown in Supplementary Fig. 13c.

Finally, we demonstrate that the ZnO stack can also be used to construct ITO-free organic photodetectors and light-emitting diodes with improved device performance. As shown in Supplementary Fig. 14a, the ZnO based organic photodetector with the active layer based on a NIR absorber (poly((4,4-dihexadecyl-4H-cyclopenta[2,1-b:3,4-b']dithiophene-2,6-diyl)-alt-[4,4'-(4,4-dihexadecyl-4H-cyclopenta[1,2-b:5,4-b'] dithio phene-2,6-diyl) bis ([1,2,5]selenadiazolo[3,4-c]pyridine)−7,7'-diyl]) (PCDTPTSe): PCBM)[72] exhibits much higher responsivity in the NIR region, compared to the ITO containing photodetectors based on the same active layer. The reason for the improved responsivity of the ZnO based photodetector could be ascribed to the lower absorption coefficient of ZnO, compared to that of ITO. This is confirmed by the optical transfer matrix model simulations, done using the real dielectric functions of the materials used in the devices[73]. As shown in Supplementary Fig. 14b, the predicted parasitic electrode absorption losses of NIR photons in the ITO-based photodetector are indeed significantly higher than those in the ZnO based device.

Also, the NIR organic light emitting diode (OLED) based on ZnO outperforms that based on ITO: As shown in Supplementary Fig. 15, the electroluminescence quantum efficiency ($EQE_{EL}$) of the ZnO based NIR OLED, with the active material based on (2,2'-((2Z,2'Z)-((12,13-bis(2-ethylhexyl)−3,9-diundecyl-12,13-dihydro[1,2,5] thiadiazolo[3,4e] thieno[2",3":4',5']thieno[2',3':4,5]pyrrolo[3,2-g]thieno[2',3':4,5] thieno[3,2-b]indole-2,10-diyl)bis(methanylylidene))bis(3-oxo-2,3-dihydro 1H-indene-2,1-diylidene))dimalononitrile) (Y5)[74], is close to 1%, about 3 times higher than that of the ITO based OLED (0.3%). The reason for the improved $EQE_{EL}$ could also be ascribed to the lower work function of ZnO, compared to ITO, leading to higher built-in electric field and reduced non-radiative surface recombination loss of charge carriers in the ZnO-based device.

## Discussion

Finding an alternative transparent electrode to ITO for the new generation of optoelectronic devices, such as organic and perovskite devices, has been challenging for the industrialization of these technologies. Tremendous efforts, based on the strategy of atomic doping or morphological modification, etc., have been put into improving the electrical conductivity and optical transparency of the transparent conducting materials, and remarkable progress has been made. Yet, few materials have been proven to be competitive to replace ITO in actual devices.

In this work, we demonstrated an alternative strategy for designing the transparent electrode for optoelectronic devices, which is based on a persistent photoinduced doping effect in sol-gel-grown ZnO films. More specifically, we demonstrated that the conductivity of the sol-gel-gown ZnO film could be increased after UV treatment, due

to oxygen vacancies trapping the UV-generated holes. Then, we developed a sequential deposition strategy, to increase the density of oxygen vacancies and the conductivity of the UV-treated ZnO. Accordingly, in the sol-gel-grown ZnO films, bearing the advantages, including scalable in both area and thickness, solution processibility, and low materials and fabrication cost, we achieved a conductivity of 460 S cm$^{-1}$, with extremely high optical transparency in both the visible and NIR spectral range. This allowed for the replacement of ITO for organic solar cells, as well as photodetectors and light emitting diodes. Also, we showed that the performance of the devices based on ZnO was improved, compared to that based on ITO. Thus, the strategy proposed in this work is expected to enable a wide usage of sequentially grown ZnO for the replacement of ITO. This is highly important for the development of industrial-compatible device architectures, and the commercialization of organic optoelectronic devices.

## Methods

### Statistics and reproducibility

All the characterizations and measurements performed in this work were repeated by different persons, and similar results were obtained.

### Experimental

Zinc acetate dihydrate (Zn(CH$_3$COO)·2H$_2$O, 99.9%), ethanolamine (NH$_2$CH$_2$CH$_2$OH, 99.5%), 2-methoxyethanol (CH$_3$OCH$_2$CH$_2$OH, Aldrich, 99.8%), and PEIE were purchased from SIGMA-ALDRICH. PBDB-T, PM6, IT4F, PCBM, and ITIC were purchased from Solarmer Materials Inc., Beijing. Y5 and Y6 were purchased from Zhiyan Company, Nanjing. ITO glass substrates were purchased from Nanbo Company, Guangzhou. Synthesis of PCDTPTSe can be found in the literature [*Chem. Mater.* 33, 5147–5155 (2021)].

The sol-gel-grown ZnO films were prepared using the method reported in the literature [*Adv. Mater.* 23, 1679 (2010)]: The precursor solution was prepared by mixing zinc acetate dihydrate (1 g), ethanolamine (277 μL), and 2-methoxyethanol (10 ml). The solution was stirred for 12 h at room temperature and then filtered using a 0.25 mm polypropylene filter, prior to use. The single-layer ZnO electrodes were deposited on clean glass substrates by spin-coating, with a speed of 4000 rpm (60 s). The substrates were cleaned in a ultrasonicator, using acetone, isopropanol, and ethanol, and then, treated by TL-1 (a mixture of NH$_3$·H$_2$O (25%):H$_2$O$_2$ (30%):H$_2$O, with a volume ratio of 1:1:5) at 85 °C for 30 min. Then, the wet ZnO films were annealed on a hotplate for 20 min. The deposition and annealing of the wet ZnO films were done in controlled environment with the relative humidity of the environment being 20%. The average thickness of the final dry ZnO films was 20–30 nm, determined by spectroscopic ellipsometry. The multi-layer ZnO stacks were prepared using the sequential deposition method: a PDMS layer was deposited to cover half of the clean glass substrate, and the ZnO layer was deposited on the other half of substrate, following the scheme shown in Fig. 4a.

The devices were constructed on ZnO or ITO-coated glass substrates. For the ITO based devices, the substrates were cleaned in a ultrasonicator, using acetone, isopropanol, and ethanol, and then, treated by TL-1 (a mixture of NH$_3$·H$_2$O (25%):H$_2$O$_2$ (30%):H$_2$O, with a volume ratio of 1:1:5) at 85 °C for 30 min. Then, a single-layer ZnO was deposited on top of ITO using the method described above. For the ZnO-based devices, the multi-layer ZnO stacks were deposited using the sequential deposition method, and a PEIE interlayer was deposited on ZnO from solution (0.04 vol%, IPA), by spin-coating with a speed of 3000 rpm (30 s), and annealed at 100 °C for 5 min. Then, the substrates were transferred into a glovebox filled with nitrogen. For constructing the solar cells, the PBDB-T:ITIC active layers were deposited from chlorobenzene solution (mixed with DIO, 0.5 vol%), with a solid concentration of 20 mg mL$^{-1}$ and a D:A weight ratio of 1:1. The solution was stirred at 80 °C for over 6 h, prior to use. The spin-coating speed used for the PBDB-T:ITIC active layers was 2200 rpm (60 s), and the

active layers were annealed at 120 °C for 10 min on a hotplate. The PM6:Y6 active layers were deposited from chloroform solution (mixed with CN, 0.5 vol%), with a solid concentration of 16 mg mL$^{-1}$ and a D:A weight ratio of 1:1.2. The solution was stirred at 40 °C for over 6 h, prior to use. The spin-coating speed used for the PM6:Y6 active layers was 2000 rpm (60 s), and the active layers were annealed at 100 °C for 10 min on a hotplate. The PM6:IT4F:PCBM active layers were deposited from chlorobenzene solution (mixed with DIO, 1 vol%), with a solid concentration of 20 mg mL$^{-1}$ and a PM6:IT4F:PCBM weight ratio of 1:1:0.2. The solution was stirred at 50 °C for over 6 h, prior to use. The spin-coating speed used for the PM6:IT4F:PCBM active layers was 2500 rpm (60 s), and the active layers were annealed at 100 °C for 10 min on a hotplate. For constructing the photodetectors, the PCDTPTSe:PCBM active layers were deposited from chlorobenzene solution (mixed with DIO, 0.5 vol%), with a solid concentration of 20 mg mL$^{-1}$ and a D:A weight ratio of 1:1. The solution was stirred at 80 °C for over 6 h, prior to use. The spin-coating speed used for the PCDTPTSe:PCBM active layers was 2200 rpm (60 s), and the active layers were annealed at 120 °C for 10 min on a hotplate. For constructing the NIR OLEDs, the Y5 active layers were deposited from chloroform solution (mixed with CN, 0.5 vol%), with a solid concentration of 15 mg mL$^{-1}$. The solution was stirred at 50 °C for over 1 h, prior to use. The spin-coating speed used for the Y5 active layers was 2000 rpm (40 s). Subsequently, a 10 nm MoO$_3$ and a 120 nm Ag layer were thermally evaporated onto the active layers through a shadow mask in a vacuum chamber at a pressure of $1 \times 10^{-6}$ mbar. The devices were then completed by the encapsulation with a glass lid, and an adhesive glue from Norland Products Inc. (NOA-73). All solar cells fabricated in this work had an active area of roughly 5 mm$^2$, determined by an optical microscope.

Thicknesses of the ZnO films were characterized using a J.A. Woolam M-2000 spectroscopic ellipsometer, and thickness of the active layers were measured by a profilometer (KLA-Tencor P-7 Stylus Profileror). Sheet resistances of the ZnO films were determined by a Keithley 2450 source meter, and a four-point probe station. AFM images of the ZnO films were measured by an atomic force microscope (MFP-3D-BIO from Oxford Instruments), using the tapping-mode. SEM images of the ZnO films were taken with Hitachi Regulus 8230. XRD measurements were performed using Bruker D8 advance. Photoluminescence measurements were carried out by using a fluorescence spectrometer (Edinburchinstruments, FS5). The excitation wavelength used was 320 nm (150 W). Absorption spectra of the active layers and transmittance of the ITO and the ZnO films were obtained by using a UV-vis-NIR spectrometer (model Lambda 950, PerkinElmer). Optical constants of the ZnO films were determined by using spectroscopic ellipsometry (J. A. Woolam M-2000). UPS and XPS measurements were performed in an ultra-high vacuum chamber with a base pressure of $10^{-10}$ mbar. A Scienta R3000 spectrometer was used for analyzing the photoelectron kinetics. UPS spectra were recorded by a He I (21.22 eV) light source with a resolution of 0.05 eV. The work function values were obtained from the secondary electron cutoff and the frontier edge of the occupied density states to vacuum level. XPS spectra were recorded using a monochromatic Al Kα 1486.6 eV as the excitation source. *J-V* curves of the solar cells were measured by a solar simulator (Newport, verasol-2, AAA) with an intensity of 100 mW cm$^{-2}$ and a Keithley 2400 source meter. The measurements were done with an optical mask (with an aperture of 2 cm$^2$) in air. The solar simulator was calibrated using a Si diode and a set of long pass filters. Light intensity-dependent *J-V* measurements were performed using the same setup used for measuring the *J-V* curves, a set of neutral density filters were used to reduce the illumination intensity of the solar simulator. EQE spectra were measured by a highly sensitive home-built setup, which consisted of a halogen lamp (LSH-75, Newport), an optical chopper, and a monochromator (CS260-RG-3-MC-A, Newport), a phase-locked amplifier (SR830, Stanford Instrument) and a current amplifier (SR570,

Stanford Instrument). A set of long pass filters (600 nm, 900 nm, 1100 nm) were used to block the overtone signals from the monochromator, and an optical aperture was used to reduce the size of the light beam from the monochromator to approximately $0.5\,mm^2$. Another halogen lamp with an intensity of $100\,mW\,cm^{-2}$ was used to provide bias illumination. EL spectra were recorded using a source meter (Keithley 2400) to inject electric current into the solar cell. A fluorescence spectrometer (KYMERA-3281-B2, Andor) with two sets of diffraction gratings, coupled to a Si EMCCD camera (DU970P-BVF, Andor) for the wavelength range of 400–1000 nm, and an InGaAs camera (DU491A-1.7, Andor) for the wavelength range of 900–1700 nm was used to collect the photons emitted from the solar cell. $EQE_{EL}$ measurements were performed using a home-built setup. A digital source meter (Keithley 2400) was used to inject electric current into the devices, and the emitted photons are collected by a Si diode. The current generated by the Si diode was measured by a picoammeters (Keithley 6482).

## Reporting summary

Further information on research design is available in the Nature Research Reporting Summary linked to this article.

## Data availability

All data supporting the findings of this study are provided within the article and the supplementary information.

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

## Acknowledgements

This work is financially supported by the National Natural Science Foundation of China (Grant No. 52073056, 51973031, and 51933001), the Fundamental Research Funds for the Central Universities (Donghua University, Grant No. 2232021A09 and 2232022A-13), the Natural Science Foundation of Shanghai (Grant No. 19ZR1401400 and 22ZR1401900). WT acknowledges funding from the European Union's Horizon 2020 research and innovation programme under grant agreement no. 851676 (ERC StGrt).

## Author contributions

Z.C. and J.W. contributed equally to this work. The project was designed by Z.T., and supervised by Z.T., W.T., and Z.M.; Z.C., J.W., and H.W. fabricated and optimized the devices and performed *J-V*,

EQE, EQE$_{EL}$, AFM, and EL measurements; J.Y. measured UPS and XPS, under the supervision of Q.B.; Y.W. and J.Z. measured PL; M.W. synthesized PCDTPTSe. Z.T., W.T., and Z.M. wrote the manuscript, and all authors contributed to the discussion and the finalizing of the manuscript.

## Competing interests

The authors declare no competing interest.
