## [Peer Review File · Nature Communications]

A Transparent Electrode Based on Solution-Processed ZnO for Organic Optoelectronic DevicesReviewers' comments:

Reviewer #1 (Remarks to the Author):

Ma et al. report the “Replacing ITO with solution processed photo-doped ZnO for organic solar cells”. They have conducted systematic study from preparation of ZnO layer on glass substrate as cathode in organic solar cells (OSCs) and properties characterization to device application. A comparison was made to the reference devices based on ITO/ZnO cathode on two blends systems, i.e., PBDB-T:ITIC and PM6:Y6. Some results are promising and attractive. However, certain contradictions remain in the device performance and stability for ZnO-based devices, which make ZnO replacing ITO not too successful as claimed in the manuscript. I am afraid it is difficult to recommend its acceptance in this journal.

1. The device stability in terms of efficiency over long-term operation and shelf storage as well as life time is important for the practical application of organic solar cells (OSCs). The development of cathodes, anodes and charge transport interlayer or even interlayer modifiers is thus challenging issues in the field of OSCs. Hence, in the development of cathodes for inverted OSCs, their light transmittance, facile preparation, device performance and stability as well as universality are crucial. In this study, photo-doped ZnO encounters decreased conductivity over storage time (Fig. S3). This will lead to decreased device performance and stability during long-term storage and operation.

2. According to literature (ca. Nat. Photon 2020, 14, 300), an efficient OSC is also a good emitter. Fig. S4 seems to tell that the ZnO-based OSCs exhibit narrower EL spectra with lower intensity over large photoresponse region. So does the absorption spectra. Can the authors comment on this?

3. As shown in Fig. 6c, the ITO-based OSCs seem to exhibit an overall larger area than ZnO-based ones, therefore the integrated current density from EQE curve for ITO-based devices should be higher than ZnO counterparts. However, the real situation is opposite.

4. When the ZnO-glass substrate used as the cathode to fabricate PM6:Y6 based OSCs, the ZnO-based devices exhibit higher JSC and VOC as well as wider and stronger EQE spectrum than ITO based ones. However, the FF is only as low as 0.42. This result at least shows the universality of ZnO replacing ITO does not exist for most higher-performance OSCs systems.

5. For PM6:Y6 system, the authors ascribed the limited PCE (9.21% vs 13.2% with ITO) as “This is primarily due to the accelerated degradation of the PM6:Y6 active layer under UV illumination”. However, they demonstrated better long-term stability for shelf storage for PM6:ITIC devices. So how about the shelf stability for PM6:Y6 devices?

6. It is suggested to provide the number of devices for data reporting (including average photovoltaic parameters and standard deviations) in this manuscript.

Reviewer #2 (Remarks to the Author):

This work by Chen et al developed a sol-gel deposited ZnO method to replace transparent ITO electrode. The method combined UV treatment, thermal annealing and multilayer deposition to get high conductivity and low optical losses. The authors demonstrated organic solar cells with the developed ZnO electrode. The work can be published. Here are my comments and questions.

1. Figure 6b, did the author UV-treat the ITO device before they measured the I-V curves? If they did not do it, I suggest the authors to do it and compare it with the ZnO device. UV treatment of the ZnO device can affect the active layer and increase the Voc.

2. The active layer that the authors used in the manuscript has a low efficiency. In the SI, the authors tried highly efficient PM6:Y6, but it did not work. The question is whether this method can be applied in a highly efficient organic solar cells? The authors should try, because there are many highly efficient organic solar cells to use.

3. Figure S8, the storage lifetime is very short for both ITO and ZnO devices. Therefore, the comparison is not very helpful. The authors can compare the device performance under continuous illumination to get more information about the stability of their ZnO electrodes.

4. The very long discussion about the Voc is not relevant to the work. The authors concluded the change of the built-in field. This is an obvious conclusion from Figure S6. Why do you need all the other discussions?

Reviewer #3 (Remarks to the Author):

The authors applied sequential deposition, annealing treatment and post UV treatment to improve the conductivity and transmittance of ZnO stack as cathode for their organic solar cell. The authors claim that solar cell performance based on their developed sol-gel-grown ZnO electrode, particularly Voc, is improved by reducing the non-radiative recombination mainly at the active layer/ZnO interface. The demonstrated Voc is one of the highest published for the selected active layer materials. However, some extra data would be required to further strengthen the claims/perspectives of this paper. I think it may be suitable for publication in the platform of Nature Communications after a major revision of the entire manuscript can be fully addressed by the authors.

1. Developing ITO-free is important to lower cost of organic solar cells/modules. However, I don't think it is the most critical element for the industrialization/commercialisation of this specific type of PV technologies as compared to further improvement of PCE, stability, lower the cost of active materials etc. So, it is a debatable statement, I would suggest authors either give a detailed technoeconomic analysis or rephrase your statement (P1 line 20-22).

2. The electrical conductivity obtained in this work (460 Scm^{-1} , P1 line 23) is an impressive result for a solution processed method. However, as it compares to sputtered ITO films ($>10^4 \text{ S cm}^{-1}$), it is far too resistive to make highly efficient opv devices without considering their long term material and resulted device stabilities. In P12, line 315-320, authors need explain clearly what the origin of the degradation of the conductivity of ZnO stack and recovery mechanism are. In my opinion, stability would be a very important element to consider the importance of their developed films and this work. In P17, line 454-458 and Fig. S8, the encapsulated ITO-based and ZnO-based devices degraded relatively fast as compared to other published OPV devices or other types of PV devices, any specific degradation mechanism for their devices?

3. In P2 line 55-57, authors cited ref 19 to support their statement. However, I found authors in ref 19 used ZnO as an interlayer rather than transparent electrode. This citation or statement need to be reconsidered.

4. Authors highlighted PEDOT:PSS, metal elements doped ZnO suffer with acidic nature led reaction with active layer materials, low NIR transmission, and poor stability. Can authors point out the issues and degradation mechanism of their photo-doped ZnO (P17, line 449-458 and Fig. S8)? Can authors answer this question along with my Question 2?

5. P4 line 92 -P5 94, would the improved band alignment by ZnO/active layer materials be system dependent? A band diagram to compare difference between ZnO and ITO would be easier for readers to understand. P7 line 192-193, it is incomplete information for readers. If it is a problem, then tell what the problem is. It requires a proper explanation and maybe a band diagram. Then it would be easier for readers to understand why you chose PEIE as your interlayer in your device fabrication.

6. If Vo is very important to increase conductivity of ZnO films, have authors tried to anneal the precursors in oxygen deficit environment such as in N₂/Ar to further enhance Vo density?

7. P6 line 164-165, I thought the conductivity of the ZnO obtained in ref 36 is higher than 100 Scm⁻¹. If so, the authors cannot make claims as highest.
8. P6 line 165-168, to reach your claim of unchanged structural and compositional properties of annealed and uv treated ZnO, I would suggest performing XPS depth profile, xrd, raman, etc. to cross-check and support your claim.
9. Fig. 2 shows AFM and XPS of ZnO annealed at different temperatures. However, Fig. 2d shows the conductivity of annealed ZnO films after UV treatment. It is confusing for readers whether Fig. 2a-2e authors applied both treatments (annealing and UV) or only heat treatment. can authors clarify it properly and make sure they show graphs with consistent experimental parameters?
10. Fig.3 The performance of single-layer ZnO based organic solar cells is limited due to low conductivity. I cannot see the significance to show this set of data in the main manuscript.
11. Fig. 3, it is not clear whether you did UV treatment only for whole multi-layer stack or each individual layer of ZnO.
12. P9 line 263-264, form the description, should the red arrow direction be in opposite direction in Fig. 4d?
13. P 11, line 282-283, if it is the case for your ZnO stack, then authors should show the experimental evidence by XRD or SEM or other techniques?
14. P11, line 295-297, if authors plotted red arrow (480nm) in Fig. 4c based on the later description and Fig. 5, authors need rearrange the order of their figures.
15. P13, line 348, can authors give the series resistance of the ZnO and ITO based opv devices?
16. P 14, line 351-353, the shapes of EQE curves can be due to the difference of the transmittance of ZnO and ITO films. Can authors show the transmittance data?
17. P14 line 353-355, the authors didn't provide the box plot of the JV characteristic. It would be more convinced for the readers if the box plot data were added.
18. Equation 3, do authors need label EQE as EQE_{pv}?
19. Can you show temperature dependent Voc curve to cross check your claim on surface recombination dominance?
20. Most leading groups could make over 18% efficient OPV device based on PM6:Y6, what is main limitations on PCE for your reference ITO and ZnO-based devices? Can you explain further?

Our responses to the reviewers' comments are provided below, and the changes made to the manuscript are marked in red.

Reviewers' comments

Reviewer #1 (Remarks to the Author):

Ma et al. report the “Replacing ITO with solution processed photo-doped ZnO for organic solar cells”. They have conducted systematical study from preparation of ZnO layer on glass substrate as cathode in organic solar cells (OSCs) and properties characterization to device application. A comparison was made to the reference devices based on ITO/ZnO cathode on two blends systems, i.e., PBDB-T:ITIC and PM6:Y6. Some results are promising and attractive. However, certain contradictions remain in the device performance and stability for ZnO-based devices, which make ZnO replacing ITO not too successful as claimed in the manuscript. I am afraid it is difficult to recommend its acceptance in this journal.

Our response: We appreciate the comments from the reviewer. In the revised version we added data from further materials systems, which confirm our claims. We, furthermore, measured the device stability, which is not reduced when replacing ITO by ZnO.

1. The device stability in terms of efficiency over long-term operation and shelf storage as well as life time is important for the practical application of organic solar cells (OSCs). The development of cathodes, anodes and charge transport interlayer or even interlayer modifiers is thus challenging issues in the field of OSCs. Hence, in the development of cathodes for inverted OSCs, their light transmittance, facile preparation, device performance and stability as well as universality are crucial. In this study, photo-doped ZnO encounters decreased conductivity over storage time (Fig. S3). This will lead to decreased device performance and stability during long-term storage and operation.

Our response: We agree with the reviewer that the device stability is crucial for the practical application of organic solar cells.

We did show that the conductivity of the UV-treated non-encapsulated ZnO decreased after long time air exposure. However, we also demonstrated that the performance of the device based on the UV-treated ZnO (encapsulated and stored in air) did not degrade so fast, because the conductivity of the ZnO electrode in the device did not decrease over time. In fact, the stability of the device based on ZnO was found to be better than that based on ITO, possibly due to the filtering of UV photons (from the ambient or induced during the JV measurements) by the thick ZnO electrode (with strong absorption band in the UV region) in the ZnO based solar cell. For the resubmission, we included the lifetime results for the ZnO solar cells based on different active materials systems stored in air, and the lifetime results for the ZnO solar cells stored with and without continuous illumination, to show that the stability of the solar cell based on ZnO is better than that based on ITO (shown below in Fig. R1 and in Supplementary Fig. 13 of the revised manuscript).

Fig. R1. Degradation of the UV-treated, encapsulated **a)** PBDB-T:ITIC and **b)** PM6:Y6 solar cells based on the ZnO stack with 6 layers of ZnO, annealed at 320 °C, compared to that of the ITO based solar cells, stored in air. **c)** Degradation of the UV-treated encapsulated PBDB-T:ITIC solar cells based on ZnO and ITO under continues solar illumination (100 mW cm^{-2}).

In addition, on Page 11-12 of the revised manuscript, we provided a discussion regarding the degradation mechanism of the UV-treated air-exposed ZnO. We ascribed the reason for the decrease of conductivity of the UV-treated air-exposed ZnO to the adsorption of oxygen at the surface of ZnO: The adsorbed oxygen molecules were converted to Zn-O bonds, by help of the free electrons in the CB of the UV-treated ZnO, resulting in the reduction of density of free electrons and Vo sites, reducing the conductivity of the UV-treated ZnO. Also, we clarified that the “recovery” of the conductivity of the UV-treated air-exposed ZnO after a second UV treatment was enabled by water adsorption: The adsorbed water reacted with the newly formed Zn-O bonds, breaking the Zn-O bonds. This led to the increased density of Vo sites in the ZnO film that could be excited again under UV illumination, allowing for the “recovery” of conductivity of the UV-treated air-exposed ZnO after a second UV treatment.

To confirm that air exposure was indeed the reason for the degradation of conductivity of the UV-treated ZnO, we investigated the degradation rate of the UV-treated encapsulated ZnO stored in a nitrogen-filled glovebox, and found that the degradation of conductivity was significantly suppressed, as shown below in Fig. R2 and in Supplementary Fig. 5b of the revised manuscript. Also, we measured the degradation of conductivity of the UV-treated ZnO stored in an oxygen-contaminated glovebox (oxygen concentration: 800 ppm, Fig. R2), we found that the degradation of conductivity of the ZnO stored in the oxygen-contaminated glovebox was much faster than that stored in the nitrogen-filled glovebox.

Fig. R2. Conductivity of ZnO stacks with 6 ZnO layers annealed at 320 °C after UV treatment (365 nm, 24 W, 600 s), stored in a nitrogen-filled glovebox and an oxygen-contaminated glovebox (oxygen concentration: 800 ppm).

As pointed out by the reviewer, the crucial factors for the development of transparent electrodes are light transmittance, facile preparation, device performance, stability, as well as universality. In the revised manuscript, we demonstrate that

- 1) **the ZnO electrode developed in this work is more transparent and easier to fabricate** (via solution processing), compared to other high-conductivity transparent electrodes,
- 2) **the ZnO based solar cells could deliver better long-term stability and performance**, compared to the ITO based solar cells,
- 3) **the ZnO electrode could also be used to replace ITO in optoelectronic devices, other than solar cells:** On Page 16 of the revised manuscript, we show that the near-infrared (NIR) photodetector based on ZnO has improved spectral responsivity, compared to that based ITO (Supplementary Fig. 14 of the revised manuscript), and the NIR organic light emitting diode based on ZnO has improved device emission quantum efficiency, compared to that based on ITO (Supplementary Fig. 15 of the revised manuscript).

These results suggest that the ZnO developed in this work is highly promising as a universal and efficient transparent electrode for optoelectronic devices in general. Thus, we believe that the revised manuscript now is able to demonstrate a successful replacement of ITO by ZnO for organic optoelectronic devices in general.

2. According to literature (ca. Nat. Photon 2020, 14, 300), an efficient OSC is also a good emitter. Fig. S4 seems to tell that the ZnO-based OSCs exhibit narrower EL spectra with lower intensity over large photoresponse region. So does the absorption spectra. Can the authors comment on this?

Our response: Fig. S4 in our previous submission is shown below (Fig. R3a). This figure shows the normalized reduced emission spectra of PBDB-T:ITIC, used only for the determination of the E_g values.

Fig. R3. a) Normalized reduced absorption and emission spectra and b) non-normalized emission spectra of the active layers of PBDB-T:ITIC in the solar cells based on the ZnO stack and ITO.

Because of the spectral normalization, we could not obtain the absolute emission intensity or the absorption strength from Fig. R3a. This is also the reason why the emission spectrum of the device based on ZnO seems to be narrower than that based on ITO. The non-normalized emission spectra of the ZnO and the ITO based devices are shown in Fig. R3b.

The emission quantum efficiency of the solar cells based on ZnO and ITO was determined by the measurement of electroluminescence external quantum efficiency (EQE_{EL}). We found that the emission efficiency of the ZnO based solar cell was about one order of magnitude higher than that of the ITO based solar cell (shown below in Fig. R4 and in Supplementary Fig. 7c of the revised manuscript), which was the reason for the higher V_{oc} , and the better performance of the ZnO based solar cell. Thus, the ZnO based solar cell is indeed the better emitter compared to the ITO based solar cell.

Fig. R4. EQE_{EL} of the solar cells based on ITO and the ZnO stack with the active layer based on PBDB-T:ITIC.

3. As shown in Fig. 6c, the ITO-based OSCs seem to exhibit an overall larger area than ZnO-based ones, therefore the integrated current density from EQE curve for ITO-based devices should be higher than ZnO counterparts. However, the real situation is opposite. Our response: We suppose that in this comment the reviewer intends to state “higher EQE” instead of “larger area”. Indeed, the J_{sc} calculated from the EQE of the solar cell based on ITO was found to be slightly higher than based on ZnO, but we claimed that the J_{sc} values, according to the JV measurements, were similar for the solar cells based

on ZnO and ITO in our previous submission. The slight difference in J_{sc} values from the JV measurements and the EQE measurements was due to the different illumination intensities used in those measurements. For the resubmission, we performed the EQE measurements with a strong bias illumination (from a halogen lamp). The intensity of the bias illumination was about 100 mW cm^{-2} , similar to the AM1.5 illumination intensity. The results are shown below in Fig. R5 and in Fig. 6d of the revised manuscript. As indicated in the figures, the J_{sc} values, calculated by integrating the product of EQE and AM1.5 spectra, are 16.85 and 16.26 mA cm^{-2} , for the solar cells based on ZnO and ITO, respectively, which agree well with the J_{sc} determined by the JV measurements.

Fig. R5. EQE spectra of the PBDB-T:ITIC solar cells based on ITO and the ZnO stack with 6 layers of ZnO, annealed at $320 \text{ }^\circ\text{C}$. The solar cell based on ZnO is UV-treated (365 nm, 24 W, 600 s), prior to characterizing its performance, and the solar cell based on ITO is not UV-treated.

4. When the ZnO-glass substrate used as the cathode to fabricate PM6:Y6 based OSCs, the ZnO-based devices exhibit higher J_{sc} and V_{oc} as well as wider and stronger EQE spectrum than ITO based ones. However, the FF is only as low as 0.42. This result at least shows the universality of ZnO replacing ITO does not exist for most higher-performance OSCs systems.

Our response: In this work, the replacement of ITO is achieved by using the UV-treated ZnO. Since the concentration of free electrons in the CB of the UV-treated non-encapsulated ZnO decreases over time, we always perform the UV treatment after the completion of device construction and encapsulation for better device performance. However, Y6 based active materials systems degrade rapidly under UV illumination [Adv. Sci. 2022, 9, 2104588]. Thus, the FFs of the UV-treated PM6:Y6 solar cells, both based on ZnO and ITO, are low (less than 50%), as shown below in Fig. R6 and in Supplementary Fig. 11 of the revised manuscript.

Fig. R6. J-V curves and the performance parameters for the UV-treated ITO based solar cells compared to the UV-treated ZnO based solar cells with the active layers based on PM6:Y6.

Therefore, the results from the PM6:Y6 systems only imply that our strategy of replacing ITO with ZnO is less efficient for the solar cells based on the active materials systems that degrade under UV illumination. Considering the fact that organic solar cells must be used under solar illumination (containing substantial UV photons), organic molecules that degrade under UV illumination would anyway not be suited for practical applications. Thus, we do not think that the universality of our strategy is compromised by the results from the PM6:Y6 based solar cells.

To address the reviewer's concern about the universality of our strategy of replacing ITO with ZnO, in the revised manuscript, we also include the experimental results from the ternary solar cells based on PM6:PCBM:IT4F, another high-efficiency materials system, found to be more stable than PM6:Y6, under UV illumination. As discussed on Page 15-16 of the revised manuscript, the performance of the PM6:PCBM:IT4F solar cell based on ZnO is more efficient than that based on ITO, due to the higher V_{oc} , similar to the results obtained from the ZnO and the ITO based solar cells based on the model PBDB-T:ITIC system. Also, on Page 16-17 of the revised manuscript, we demonstrate that the device performance of NIR organic photodetectors, as well as NIR OLEDs could be improved by using ZnO instead of ITO as the transparent electrode.

Now with the added experiments, and the substantial revision of the manuscript, our method to replace ITO is found to be effective for solar cells based on different active materials systems, and also for photodetectors and OLEDs. Thus, we believe that the universality of our strategy of replacing ITO with ZnO is well confirmed.

5. For PM6:Y6 system, the authors ascribed the limited PCE (9.21% vs 13.2% with ITO) as "This is primarily due to the accelerated degradation of the PM6:Y6 active layer under UV illumination". However, they demonstrated better long-term stability for shelf storage for PM6:ITIC devices. So how about the shelf stability for PM6:Y6 devices?

Our response: We suppose that in this comment the reviewer intends to state "PBDB-T:ITIC" instead of "PM6:ITIC". Indeed, the PM6:Y6 system degraded faster than the PBDB-T:ITIC system, because of the more severe UV stability issue of Y6, compared to that of ITIC [Adv. Sci. 2022, 9, 2104588].

For the resubmission, we tested the shelf stability of the PM6:Y6 devices, as shown

in Fig. R1b and in Supplementary Fig. 13b of the revised manuscript. The shelf stability of the ZnO device based on PM6:Y6 is worse than that based on PBDB-T:ITIC. Nevertheless, the shelf stability of the PM6:Y6 device based on ZnO is better than that based on ITO, possibly due to the filtering of UV photons by the thick ZnO electrode in the ZnO based device. Furthermore, we show that under continuous illumination, the ZnO based solar cell is more stable than the ITO based solar cell (Fig. R1c and Supplementary Fig. 13c of the revised manuscript).

6. It is suggested to provide the number of devices for data reporting (including average photovoltaic parameters and standard deviations) in this manuscript.

Our response: We included statistic results for the photovoltaic parameters of the solar cells in Supplementary Table 1 of the revised manuscript. For each materials system, the average values and the errors are derived from 10 devices fabricated using the same processing condition.

Reviewer #2 (Remarks to the Author):

This work by Chen et al developed a sol-gel deposited ZnO method to replace transparent ITO electrode. The method combined UV treatment, thermal annealing and multilayer deposition to get high conductivity and low optical losses. The authors demonstrated organic solar cells with the developed ZnO electrode. The work can be published. Here are my comments and questions.

Our response: We thank the reviewer for the positive comments.

1. Figure 6b, did the author UV-treat the ITO device before they measured the I-V curves? If they did not do it, I suggest the authors to do it and compare it with the ZnO device. UV treatment of the ZnO device can affect the active layer and increase the Voc.

Our response: We did not UV treat the ITO device before the JV measurements. For the resubmission, we included the data for the UV-treated ITO based device, and compared the performance of the UV-treated ITO based and the UV-treated ZnO based devices.

As shown in Supplementary Fig. 6 of the revised manuscript, the FF and Voc of the ITO based PBDB-T:ITIC device are slightly reduced by the UV treatment, thus, the overall performance of the UV-treated PBDB-T:ITIC device based on ITO is still worse than that based on ZnO. For the PM6:Y6 solar cells based on ITO, the device performance is much worse after UV illumination (Supplementary Fig. 11 of the revised manuscript), due to the degradation of the Y6 molecules in the active layer [*Adv. Sci.* 2022, 9, 2104588]. Thus, the PCE of the UV-treated PM6:Y6 solar cell based on ITO is found to be lower than that based ZnO.

2. The active layer that the authors used in the manuscript has a low efficiency. In the SI, the authors tried highly efficient PM6:Y6, but it did not work. The question is whether this method can be applied in a highly efficient organic solar cells? The authors should try, because there are many highly efficient organic solar cells to use.

Our response: The ZnO based PM6:Y6 solar cell has poor performance because Y6 degrades under UV illumination [*Adv. Sci.* 2022, 9, 2104588]. For the resubmission, we constructed high-efficiency ternary solar cells, using the active material system of

PM6:IT4F:PCBM, which is found to be more stable than the Y6 based materials systems under UV illumination: High efficiency (over 12%) is realized using ZnO as the transparent electrode of the PM6:IT4F:PCBM solar cell, and also, the Voc of the solar cell based on ZnO is found to be higher than that based ITO. The results are provided in Supplementary Fig. 12 of the revised manuscript.

3. Figure S8, the storage lifetime is very short for both ITO and ZnO devices. Therefore, the comparison is not very helpful. The authors can compare the device performance under continuous illumination to get more information about the stability of their ZnO electrodes.

Our response: For the resubmission, we performed stability test for the solar cells under continuous solar illumination (100 mW cm^{-2}), and the results are provided in Supplementary Fig. 13c of the revised manuscript: Under continuous illumination, the stability of the solar cell based on ZnO is better than that based on ITO, possibly due to the filtering of UV photons by the thick ZnO electrode in the ZnO based solar cell.

4. The very long discussion about the Voc is not relevant to the work. The authors concluded the change of the built-in field. This is an obvious conclusion from Figure S6. Why do you need all the other discussions?

Our response: Currently, the major limit for the performance of organic solar cells is the high voltage loss, thus, we believe that the observed improvement in Voc of the ZnO based solar cell is a very inspiring result. However, some of the readers may not be very familiar with the dependence of Voc on the built-in field, and would likely be skeptical about whether there are other possible reasons for the improved Voc. Thus, we provided a systematic analysis on the voltage losses in the devices based ZnO and ITO, to show that the increased built-in field was indeed the reason for the improved Voc. In the revised manuscript, we moved the majority part of the discussion about Voc to Supplementary Fig. 7.

Reviewer #3 (Remarks to the Author):

The authors applied sequential deposition, annealing treatment and post UV treatment to improve the conductivity and transmittance of ZnO stack as cathode for their organic solar cell. The authors claim that solar cell performance based on their developed sol-gel-grown ZnO electrode, particularly Voc, is improved by reducing the non-radiative recombination mainly at the active layer/ZnO interface. The demonstrated Voc is one of the highest published for the selected active layer materials. However, some extra data would be required to further strengthen the claims/perspectives of this paper. I think it may be suitable for publication in the platform of Nature Communications after a major revision of the entire manuscript can be fully addressed by the authors.

Our response: We thank the reviewer for the overall positive comments.

1. Developing ITO-free is important to lower cost of organic solar cells/modules. However, I don't think it is the most critical element for the industrialization/commercialisation of this specific type of PV technologies as compared to further improvement of PCE, stability, lower the cost of active materials etc. So, it is a debatable statement, I would suggest authors either give a detailed techno-economic analysis or rephrase your statement (P1

line 20-22).

Our response: We agree with the reviewer that further improving the PCE, the device stability, and lowering the cost of active materials are also highly important for the industrialization of the OPV technology. It is indeed debatable whether some of these tasks are more important than the others. We do not intend to claim that replacing ITO is the most critical task, thus, we rephrased the statement quoted by the reviewer for the revised manuscript.

2. The electrical conductivity obtained in this work (460 S cm^{-1} , P1 line 23) is an impressive result for a solution processed method. However, as it compares to sputtered ITO films ($>10^4 \text{ S cm}^{-1}$), it is far too resistive to make highly efficient opv devices without considering their long term material and resulted device stabilities. In P12, line 315-320, authors need explain clearly what the origin of the degradation of the conductivity of ZnO stack and recovery mechanism are. In my opinion, stability would be a very important element to consider the importance of their developed films and this work.

Our response: For the resubmission, we provided more systematic analyses on the degradation of conductivity of the UV-treated ZnO, stored in air. Our explanation regarding the degradation and recovery is provided below and on Page 11-12 of the revised manuscript:

The decrease of conductivity of n-type ZnO (with free electrons in CB) stored in air is not unexpected, which can be ascribed to oxygen adsorption on the surface of ZnO. In the literature [*Physica E Low Dimens. Syst. Nanostruct.* 2013, 48, 7], Qiao et al. reported that the hollow sites at the hexagon centers of the ZnO surface were capable of adsorbing oxygen molecules. Thus, after the adsorption of oxygen molecules, there was electron transfer from the ZnO surface to the adsorbed oxygen. This led to the formation of Zn-O bonds, the reduction of density of free electrons and Vo sites, and the increase of the resistance of the ZnO film. In this work, we employ UV treatment to substantially increase the electron concentration in the CB, and the conductivity of ZnO. Thus, exposing the UV-treated ZnO to air is expected to lead to the reduction of electron concentration in ZnO, and the reduction of ZnO conductivity.

Meanwhile, exposing the UV-treated ZnO to air also leads to additional water adsorption. The adsorbed water is expected to react with the newly formed Zn-O bonds, breaking the Zn-O bonds, as discussed in the manuscript. This gives rise to the increase of the density of Vo sites in the ZnO, which could be excited again under UV illumination. This is believed to be the reason for the “recovery” of conductivity of the UV-treated air-exposed ZnO after a second UV treatment.

To confirm that air exposure is indeed the reason for the degradation of conductivity of the UV-treated ZnO, we investigated the degradation rate of the UV-treated encapsulated ZnO stored in a nitrogen-filled glovebox, and found that the degradation of conductivity was significantly suppressed (Supplementary Fig. 5b of the revised manuscript). Also, we studied the degradation of conductivity of the UV-treated ZnO stored in an oxygen-contaminated glovebox (oxygen concentration: 800 ppm), and found that the degradation of conductivity is much faster than that of the ZnO stored in

a nitrogen-filled glovebox (Supplementary Fig. 5b of the revised manuscript).

Thus, we are not so concerned about the degradation of the UV-treated ZnO due to air exposure. Because we could perform the UV treatment after the completion of device construction and encapsulation. In this case, we could not find any indication suggesting that the use of the UV-treated ZnO would cause any stability issue for the solar cells. In fact, we found that the stability of the solar cell based on ZnO is generally better than that based on ITO, possibly due to the filtering of UV photons (from the ambient or induced during the JV measurements) by the thick ZnO electrode (with strong absorption band in the UV region) in the ZnO based solar cell, as shown in Supplementary Fig. 13 of the revised manuscript.

In P17, line 454-458 and Fig. S8, the encapsulated ITO-based and ZnO-based devices degraded relatively fast as compared to other published OPV devices or other types of PV devices, any specific degradation mechanism for their devices?

Our response: We agree with the reviewer that the degradation rates of the devices studied in this work were higher than that reported in the literature focusing on the investigation of lifetime of organic solar cells. Generally, the degradation of an organic solar cell is accelerated by the oxygen and water molecules diffusing into the active layer of the device [*Sol. Energy Mater. Sol. Cells* 2011, 95, 1268]. Therefore, the encapsulation method used to prevent the entering of the oxygen and water molecules into the device is critically important in determining the stability of the solar cell. For realizing the most stable organic solar cell, a special encapsulation glue and often a special moisture absorber needs to be employed, and the geometry of the encapsulation lid needs to be carefully optimized [*Adv. Funct. Mater.* 2021, 31, 2100151].

However, the access to the detailed information of the encapsulation method used by the specialized labs is often limited, and we have to admit that we are not very experienced in finding the most effective method to encapsulate organic solar cells. In this work, we encapsulated our devices using a glass lid, and an adhesive glue from Norland Products Inc. (NOA-73). We could note that the lifetime of the device after encapsulation was significantly improved, from a few hours to a few weeks. However, our method of encapsulation is clearly less effective, compared to that used in the literature from the specialized research labs.

3. In P2 line 55-57, authors cited ref 19 to support their statement. However, I found authors in ref 19 used ZnO as an interlayer rather than transparent electrode. This citation or statement need to be reconsidered.

Our response: Indeed, reference 19 used in our previous submission was about using ZnO as the interlayer. We intended to use it as the reference to stress that ZnO was highly promising due to its solution processability. In the revised manuscript, more references are added for the use of ZnO as the electrode for organic solar cells.

4. Authors highlighted PEDOT:PSS, metal elements doped ZnO suffer with acidic nature led reaction with active layer materials, low NIR transmission, and poor stability. Can authors point out the issues and degradation mechanism of their photo-doped ZnO (P17,

line 449-458 and Fig. S8)? Can authors answer this question along with my Question 2?
Our response: We have provided detailed discussions on the degradation and recovery mechanism of the UV-treated ZnO in the revised manuscript, and in our detailed response to Question 2 from the reviewer.

5. P4 line 92 -P5 94, would the improved band alignment by ZnO/active layer materials be system dependent? A band diagram to compare difference between ZnO and ITO would be easier for readers to understand. P7 line 192-193, it is incomplete information for readers. If it is a problem, then tell what the problem is. It requires a proper explanation and maybe a band diagram. Then it would be easier for readers to understand why you chose PEIE as your interlayer in your device fabrication.

Our response: Yes, the band alignment is dependent on the active material selection: The band alignment at the cathode interface of organic solar cells depends on the work function (WF) of the cathode and the LUMO of the acceptor material (responsible for electron transport) in the blend active layer [*Adv. Energy Mater.* 2014, 4, 1400643]. An explanation for the need to use PEIE for the ZnO based solar cells is provided below:

For an organic solar cell, ohmic contacts are needed to realize the high built-in electric field, facilitate charge carrier extraction, and thus achieve good device performance. To realize ohmic electron contact at the cathode/active layer interface of an organic solar cell, the work function of the cathode should be lower than the energy of the LUMO of the acceptor. Typically, the LUMO energy level of the acceptor in an organic solar cell is at about 3.9-4.2 eV. Therefore, the work function of the ZnO film (4.0-4.1 eV) developed in this work could be slightly too high to allow for the formation of ohmic electron contact. PEIE is frequently used to lower the work function of an electrode layer [*Science* 2012, 336, 327], and it also works for the ZnO electrode.

To address the Question 10 from the reviewer, and to reduce the length of the revised manuscript, we removed the discussion about the solar cells based on the single-layer ZnO electrode and the problem of using the single-layer ZnO as the cathode for organic solar cells. Nevertheless, in the revised manuscript, we provided a schematic picture for the band alignment and an explanation for the use of PEIE for the ZnO stack based organic solar cell, as shown in Fig. 5a and Supplementary Fig. 9 of the revised manuscript, and below in Fig. R7.

Fig. R7. a) UPS spectra of ITO and the ZnO stack with 6 layers of ZnO, annealed at 320 °C, **b)** UPS spectra of ITO coated with a single-layer ZnO and the ZnO stack modified by PEIE. **c)** Band diagram for the cathode interfaces in the PBDB-T:ITIC solar cells based on ZnO, PEIE modified ZnO, and ITO/ZnO.

6. If V_o is very important to increase conductivity of ZnO films, have authors tried to anneal the precursors in oxygen deficit environment such as in N_2/Ar to further enhance V_o density?

Our response: We did try to anneal the precursor in a nitrogen-filled glovebox, but we found that the conductivity of the film processed in an inert atmosphere was extremely low (beyond the measurement range of the setup used for determining conductivity). This is because of the need to have water to participate in the formation of ZnO nanocrystals using the sol-gel method [Appl. Surf. Sci. 2020, 512, 145660].

More specifically, in the sol-gel grown film, we have ZnO nanocrystals as well as zinc acetate that are not decomposed. The presence of zinc acetate increases the resistance of the sol-gel grown ZnO film. Therefore, to grow a high-conductivity ZnO, we first need to make sure that the degree of formation of ZnO nanocrystals is high, then we need to induce a high density of V_o sites in the ZnO nanocrystals for the high-efficiency UV-doping.

However, the degree of formation of ZnO nanocrystals depends on the hydrolysis kinetics of the precursor film, which is strongly dependent on the environmental humidity. In the reference [Appl. Surf. Sci. 2020, 512, 145660], it was demonstrated that both too-low and too-high relative humidity could lead to undesired hydrolysis kinetics, and thus, incomplete formation of ZnO nanocrystals in the sol-gel grown ZnO film. Therefore, the desired relative humidity for processing high-conductivity sol-gel ZnO was found to be

25%, which is similar to the humidity condition used in this work (20%).

Reducing oxygen level to realize increased density of V_o in the sol-gel grown ZnO is thus practically very challenging, because it would require a well control over both the oxygen and the water concentration. This high challenge is in fact the motivation for us to develop the sequential deposition strategy.

7. P6 line 164-165, I thought the conductivity of the ZnO obtained in ref 36 is higher than 100 Scm^{-1} . If so, the authors cannot make claims as highest.

Our response: After re-evaluating the results reported in the reference, we found that indeed a conductivity value over 100 S cm^{-1} was realized in that reference. Thus, we rephrased our statement in the revised manuscript. However, it is worth to note that the conductivity reported in that reference is still significantly lower than the conductivity of the sequentially deposited ZnO (over 400 S cm^{-1}) developed in this work.

8. P6 line 165-168, to reach your claim of unchanged structural and compositional properties of annealed and uv treated ZnO, I would suggest performing XPS depth profile, xrd, raman, etc. to cross-check and support your claim.

Our response: For the resubmission, we provided XRD and SEM results for the UV-treated and untreated ZnO films annealed with different temperatures. As shown in Supplementary Fig. 2 and Fig. 3 of the revised manuscript, the XRD and SEM results do support our claim of unchanged structural and compositional properties of ZnO after the UV treatment.

9. Fig. 2 shows AFM and XPS of ZnO annealed at different temperatures. However, Fig. 2d shows the conductivity of annealed ZnO films after UV treatment. It is confusing for readers whether Fig. 2a-2e authors applied both treatments (annealing and UV) or only heat treatment. can authors clarify it properly and make sure they show graphs with consistent experimental parameters?

Our response: To avoid confusion, we split the figure into two separated figures (Fig. 2 and Fig 3), and clarified the experimental conditions in the captions of those figures in the revised manuscript.

10. Fig.3 The performance of single-layer ZnO based organic solar cells is limited due to low conductivity. I cannot see the significance to show this set of data in the main manuscript.

Our response: We removed the discussion about the single-layer ZnO based solar cells from the revised manuscript.

11. Fig. 3, it is not clear whether you did UV treatment only for whole multi-layer stack or each individual layer of ZnO.

Our response: For the conductivity measurements, we did the UV treatment for the whole multi-layer ZnO stack. To realize high performance ZnO based solar cells, we performed the UV treatment after the completion of device construction and encapsulation. This is better clarified in the revised manuscript.

12. P9 line 263-264, from the description, should the red arrow direction be in opposite direction in Fig. 4d?

Our response: We suppose that the reviewer is referring to Fig. 4c in our previous submission. In that figure, the blue arrow, pointing down, is the arrow to indicate the transition from CB to Vo, and the red arrow, pointing up, is the arrow to indicate the transition from Vo to CB. We now understand that the inclusion of the red arrow could lead to confusion. Thus, we removed the red arrow from the figure, and use the figure only to illustrate the decay kinetics of charge carriers, as shown below in Fig. R8 and in Fig. 4c of the revised manuscript.

Fig. R8. Energy diagram for the different transitions in the ZnO stack, identified by the PL measurements.

13. P 11, line 282-283, if it is the case for your ZnO stack, then authors should show the experimental evidence by XRD or SEM or other techniques?

Our response: we performed XRD and SEM measurements for the ZnO stack with different numbers of ZnO layers for the resubmission, as shown in Supplementary Fig. 4 of the revised manuscript. From the SEM images, we observe that the size of the ZnO nanocrystals reduces with the increasing number of ZnO layers in the stack, and from the XRD spectra, we find that the intensity of the diffraction peaks reduces with the increasing number of ZnO layer in the stack. These results suggest that the repeated intensive thermal treatments for growing the high temperature annealed ZnO stack lead to a structural change of the ZnO films in the stack.

14. P11, line 295-297, if authors plotted red arrow (480nm) in Fig. 4c based on the later description and Fig. 5, authors need rearrange the order of their figures.

Our response: We removed the red arrow from Fig. 4c to avoid confusion, and revised the words quoted by the reviewer.

15. P13, line 348, can authors give the series resistance of the ZnO and ITO based opv devices?

Our response: We performed dark IV measurements for the ZnO and the ITO based solar cells for the resubmission, and calculated the device series resistance from the dark IV curves, as shown below in Fig. R9 and in Fig. 6c of the revised manuscript.

Fig. R9. Dark I-V curves of the PBDB-T:ITIC solar cells based on ITO and the ZnO stack with 6 layers of ZnO, annealed at 320 °C. The solar cell based on ZnO is UV-treated (365 nm, 24 W, 600 s), prior to characterizing its performance, and the solar cell based on ITO is not UV-treated.

16. P 14, line 351-353, the shapes of EQE curves can be due to the difference of the transmittance of ZnO and ITO films. Can authors show the transmittance data?

Our response: We ascribed the reason for the different spectral shapes of EQE to the different optical constants of ZnO and ITO. This means that the transmittance spectra of the ZnO and the ITO film must be different. In Fig. 5e of the revised manuscript and in Fig. R10 shown below, the transmittance spectra are provided. Nevertheless, we should note that the shape of EQE spectrum is also determined by the optical interference condition in the solar cell. A higher transmittance of the ZnO stack, compared to that of ITO, at a specific wavelength, does not necessarily lead to a higher EQE value, because of the possible higher reflection or metal absorption losses at that wavelength. However, generally, an overall more transparent electrode would lead to lower optical losses in the device when the optical interference condition in the device is optimized.

Fig. R10. Transmittance spectra of the ZnO stack (150 nm) and the ITO film (170 nm) used in this work.

17. P14 line 353-355, the authors didn't provide the box plot of the JV characteristic. It would be more convinced for the readers if the box plot data were added.

Our response: We included statistic results for the photovoltaic parameters of the solar cells in Supplementary Table 1 of the revised manuscript. The average values and the errors are derived from 10 devices fabricated using the same processing condition.

18. Equation 3, do authors need label EQE as EQE_{pv}?

Our response: In the revised manuscript, we use EQE as the abbreviation for external quantum efficiency.

19. Can you show temperature dependent Voc curve to cross check your claim on surface recombination dominance?

Our response: We currently do not have the access to the temperature dependent measurement, however, to the best of our knowledge, temperature dependent Voc is influenced by many factors and it is not straight-forward to deduce the dominance of surface recombination, in particular in organic solar cells, where charge separation depends on the temperature as well. In this work, the enhanced Vbi deduced from the UPS measurements is strong evidence that the changes in Voc are related to changes in Vbi. This already implies that Vbi affects Voc, and thus recombination at the interfaces with the charge transport layers. This is also consistent with the fact that Voc enhancement was observed for various active material systems.

20. Most leading groups could make over 18% efficient OPV device based on PM6:Y6, what is main limitations on PCE for your reference ITO and ZnO-based devices? Can you explain further?

Our response: To the best of our knowledge, the highest efficiency values are realized using the lab synthesized active materials, and via molecular structural fine tuning of the donor and the acceptor material. Since we were not specialized in organic materials synthesis, we relied on the use of the commercially available materials for this study.

Nevertheless, the highest PCE value for the solar cells based on PM6:Y6 realized in this work is 13.2%, which we think is not really much worse than that first reported in the literature (15.7%) [*Joule* 2019, 3, 1140]. We speculate that the reason for the relatively lower PCE of the solar cell constructed in this work is the undesired molecular weight of the commercially obtained donor PM6, since the performance of the solar cells based on PM6:Y6 is known to be strongly dependent on the molecular weight of the donor polymer [*Adv. Energy Mater.* 2021, 11, 2002709].

REVIEWERS' COMMENTS

Reviewer #1 (Remarks to the Author):

The authors have revised the manuscript according to the reviewer comments. It can be accepted now.

Reviewer #2 (Remarks to the Author):

The authors have addressed all my technique questions.

Reviewer #3 (Remarks to the Author):

The authors have satisfactorily addressed all my comments and improved the quality of paper in the revised version of the manuscript. I would like to recommend accepting the paper for publication in Nature Communications.